# PIKI-1, a class II PI 3-kinase, functions in endocytic trafficking

Gabrielle R. Reimann[1], Philip T. Edeen[1], Sylvia Conquest[1], Barth D. Grant[2], David S. Fay[1]*

**1** Department of Molecular Biology, College of Agriculture Life Sciences, and Natural Resources, University of Wyoming, Laramie, Wyoming, United States of America, **2** Department of Molecular Biology and Biochemistry, Rutgers University, Piscataway, New Jersey, United States of America

* davidfay@uwyo.edu

## Abstract

Cellular membrane trafficking, including endocytosis and exocytosis, is a complex process coordinated by trafficking-associated proteins, cargo molecules, the cytoskeleton, and membrane lipids. The NIMA-related kinases NEKL-2 (human NEK8/9) and NEKL-3 (human NEK6/7) are conserved regulators of membrane trafficking in *Caenorhabditis elegans* that are required for the completion of molting. Using a genetic approach, we isolated reduction-of-function mutations in *piki-1* that suppress *nekl*-associated molting defects. *piki-1* encodes the sole predicted *C. elegans* Class II PI 3-kinase (PI3K), a relatively understudied class of lipid modifiers that contribute to the production of PI 3-phosphate (PI(3)P) and PI 3,4-bisphosphate (PI(3,4)$P_2$). Using genetically encoded lipid sensors, we found that PIKI-1 was responsible for the production of PI(3,4)$P_2$ in the *C. elegans* epidermis but played only a minor role in contributing to PI(3)P levels. Consistent with this, both PI(3,4)$P_2$ and PIKI-1 partially colocalized to early endosomes, and reduction of PIKI-1 affected the size and protein composition of early endosomal compartments marked by RAB-5, EEA-1, and SNX-1. Reduced PIKI-1 also led to increased tubulation of endosomal compartments associated with recycling or the degradation of cellular debris. In contrast to studies using mammalian cell culture, PIKI-1 was largely dispensable for clathrin-mediated endocytosis in the worm epidermis, a polarized epithelium. Notably, reduction of PIKI-1 function mitigated defects in early endosomes associated with the depletion of NEKL-2. We propose that reduction of PIKI-1 function may suppress *nekl* molting defects by partially restoring endocytic trafficking function within a subset of compartments, including the early endosome. We also show that inhibition of HIPR-1, an ortholog of the mammalian PI(3,4)$P_2$-binding proteins, HIP1 and HIPR1, suppresses *nekl* molting defects, consistent with a model that loss of PIKI-1 alters the binding of endocytic regulators in a manner that partially compensates for the loss of NEKL-2 activity.

**Data availability statement:** All relevant data are in the manuscript and its supporting information files.

**Funding:** o This work was funded by National Institutes of Health (NIGMS) grants GM136236 and GM103432 to DSF and GM135326 to BDG. This included salary for DSF (GM136236 and GM103432), PTE (GM136236), GRR (GM136236), SC (GM136236), and BDG (GM135326). The funders had no role in study design, data collection and analysis, decision to publish, or preparation of the manuscript.

**Competing interests:** The authors have declared that no competing interests exist.

## Author summary

The uptake of materials from outside the cell and their subsequent delivery to specific intracellular locations are essential for cell function and survival. Two of the mechanisms that control this complex intracellular pathway involve the modification of proteins and of lipids, processes that are highly conserved across species. In this study, we used the model organism *Caenorhabditis elegans*, which is highly amenable to cell biological and genetic approaches, to establish a novel connection between these two regulatory mechanisms and demonstrate the importance of lipid modifications in maintaining the normal functioning of intracellular transport. Our results also provide insights into the fundamental cellular functions of proteins associated with human disease including cancer and metabolic disease.

## Introduction

Polyphosphoinositides (PPIns) are a family of small phospholipids embedded in the cytosolic leaflet of the plasma membrane and membrane-bound organelles [1–4]. PPIns are derived from the membrane lipid phosphatidylinositol (PI), which consists of a glycerol backbone esterified to two fatty acid chains and a phosphate group that links to a polar myo-inositol head group projecting into the cytoplasm (Fig 1A). Modification of the *myo*-inositol head group at hydroxyl positions 3', 4', and 5' can lead to the production of seven different PPIns species (Fig 1B). Interconversion of PI and PPIns species is controlled by several classes of PI kinases and phosphatases (Fig 1B) [5]. The interconnected nature of the PPIns pathway and the ability of PPIns to undergo reversible modifications allows for tight spatiotemporal control of PPIns on cellular membranes [5–7]. Correspondingly, the concentration of specific PPIns on cytosolic leaflets controls the recruitment of membrane trafficking and signaling proteins, thereby impacting fundamental cellular processes [4–7]. Although crucial for regulation, PPIns are a low abundance lipid species; PI makes up ~10% of the total membrane phospholipid pool whereas PPIns are estimated to comprise only 0.2–1% and are comparatively short-lived [2,8,9].

One family of lipid modifiers, phosphatidylinositol 3-kinases (PI3Ks), specifically phosphorylate the hydroxyl group at position 3' on the inositol ring to produce three different PPIns species; PI (3,4,5)-trisphosphate [PI(3,4,5)P$_3$], PI 3,4-bisphosphate [PI(3,4)P$_2$], and PI 3-phosphate [PI(3)P] (s 1B) [10–12]. PI3Ks are subdivided into three classes, based on the PPIns species they produce, their protein domains, and their interactions with different regulatory subunits [12–14]. Class I PI3Ks produce PI(3,4,5)P$_3$ from PI(4,5)P$_2$ and are involved in pathways that regulate cell growth, proliferation, metabolism, and autophagy [15–17]. Class III PI3Ks produce PI(3)P from PI and primarily regulate membrane trafficking, including endosome-to-lysosome maturation and autophagy [16,18]. Class II PI3Ks can produce PI(3)P from PI as well as PI(3,4)P$_2$ from PI(4)P and are less well characterized than Class I or III PI3Ks

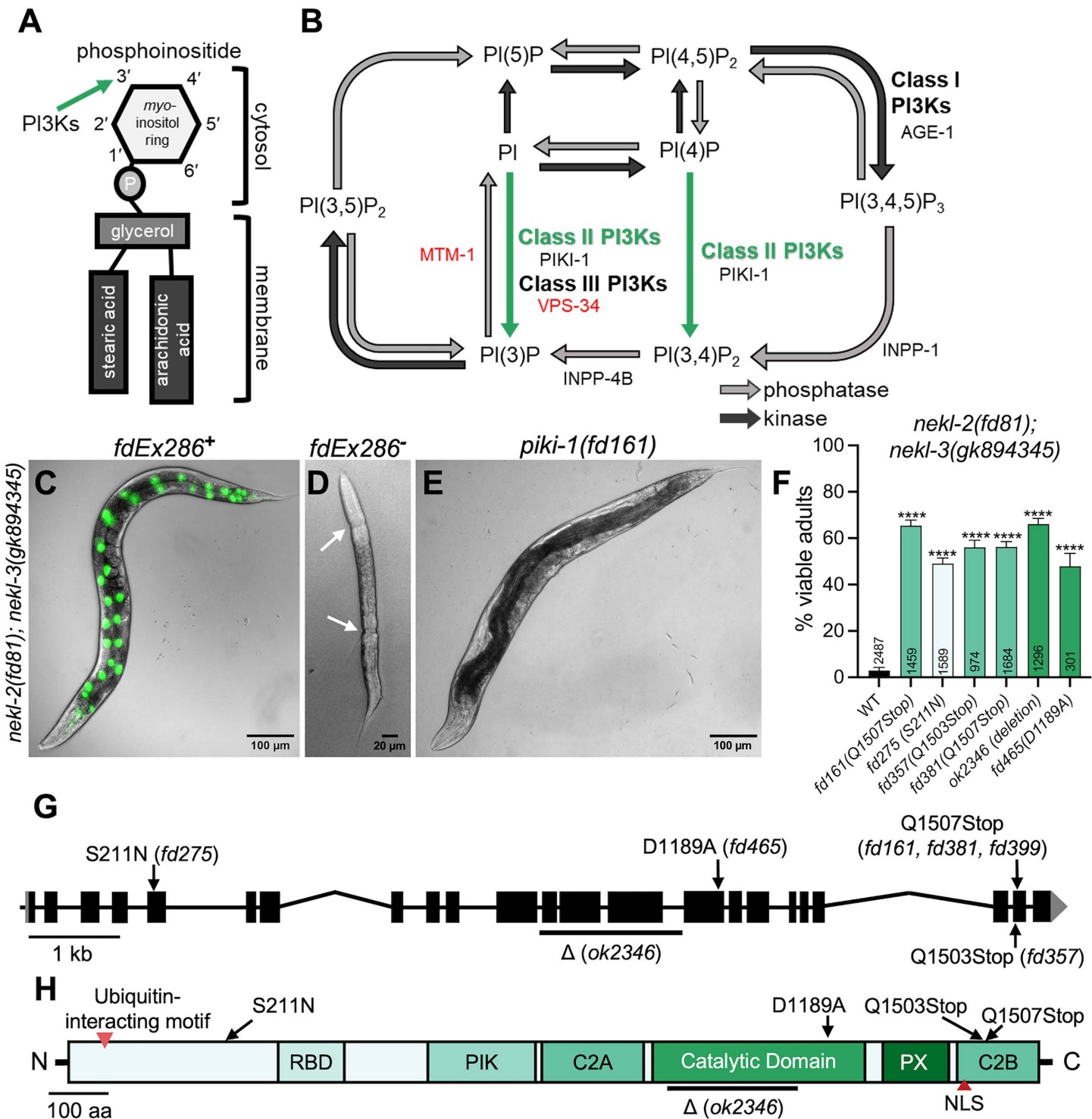

**Fig 1. *nekl*-associated molting defects are suppressed by loss of function of *piki-1*.** (A) Graphical representation of a phosphatidylinositol (PI). The 3' position in the inositol ring that is phosphorylated by PI 3-kinases (PI3Ks) is indicated by the green arrow. (B) Schematic diagram of the PPIns biosynthesis pathway. Dark gray arrows represent lipid kinase activity; light gray arrows indicate lipid phosphatase activity. The reactions catalyzed by the different classes of PI3Ks are indicated. Green arrows indicate reactions catalyzed by class II PI3Ks. *C. elegans* genes coding for kinases and phosphatases are listed by the reaction they are predicted catalyze; essential genes are in red text. (C) Merged DIC and fluorescence image of a

representative *nekl-2(fd81); nekl-3(gk894345)* worm carrying an extrachromosomal rescuing array (*fdEx286*) containing wild-type *nekl-3* and a broadly expressed reporter (SUR-5::GFP). (D) DIC image of an L2/L3 arrested *nekl-2(fd81); nekl-3(gk894345)* larva that failed to inherit *fdEx286*. White arrows indicate boundaries where the old cuticle has been shed from the head and tail but not from the midbody. (E) Representative DIC image of a suppressed *nekl-2(fd81); nekl-3(gk8942345); piki-1(fd161)* mutant, which does not require *fdEx286* for viability. (F) Percentages of viable adults for the indicated genotypes; corresponding amino acid changes are indicated in parentheses next to the *piki-1* allele designation. The number of animals analyzed per genotype is indicated in the graph. Data are shown as the mean and 95% confidence interval (CI). Statistical significance was determined using Fischer's exact test; ****$p \leq 0.0001$ relative to control (WY1145). Raw data are available in S1 File. (G) Schematic diagram of the *piki-1* gene; black rectangles denote exons and lines denote introns. The locations of missense alleles are indicated by arrows. The large deletion (Δ) *ok2346* is indicated by a line beneath the affected exons and introns. (H) Schematic diagram of the PIKI-1 protein showing predicted domains; red arrowheads indicate known motifs. Black arrows indicate the positions of amino acids altered in *piki-1* mutants; black line indicates the region affected by the *ok2346* deletion. C2A, first C2 domain; C2B, second C2 domain; [NLS, nuclear localization signal;] PIK, PI3K class II accessory domain; PX, phox homology domain; RBD, Ras-binding domain.

(Fig 1B) [19–21]. This is due in part to the partial functional overlap of Class II PI3Ks with both Class I and III enzymes [12,14,16,22]. For example, both Class II and Class III enzymes can produce PI(3)P [7,19,21,23–25], and PI(3,4)P$_2$ can be generated by the 5' dephosphorylation of the Class I PI3K product, PI(3,4,5)P$_3$ (Fig 1B) [26–31]. Class II PI3Ks also differ from Class I and III enzymes in that they are not known to form stable complexes with regulatory units [20,21,32–36]. The regulatory subunits of Class I and III PI3Ks have been shown to control the localization and activity of these kinases whereas the regulation of Class II PI3Ks is less well understood. Finally, the connection of Class II PI3Ks to human disease is less well established than for class I and III enzymes [11,37,38].

The role of PI3Ks and their products is crucial for multiple steps in membrane trafficking [2,3,39]. For example, uptake of cargo by clathrin-mediated endocytosis is dependent on PI 4,5-bisphosphate [PI(4,5)P$_2$], which is required for initiating formation of the clathrin lattice on budding vesicles [40–45]. During maturation of the clathrin-coated pit, other PI3Ks and lipid phosphatases are subsequently recruited to modify the PPIns population, which is necessary for the recruitment of proteins that promote vesicle scission and the internalization of cargo [46]. After internalization, nascent vesicles fuse with the early endosome (also referred to as the sorting endosome), where cargo is sorted for delivery to specific intracellular locations [47,48]. Within the early endosome, the PPIns population consists primarily of PI(3)P, produced by class III PI3Ks with some contribution from class II PI3Ks [49,50]. Here, PI(3)P recruits early endosome effectors necessary for vesicular docking, fusion, and the subsequent sorting of cargo [7,49–52]. During maturation of the early endosome additional subdomains develop, a process that is driven by cargo, membrane-associated proteins, and underlying changes in the composition of the PPIns [49]. For example, PI(3,5)P$_2$ affects the sorting of proteins destined for late endosomes and degradation in the lysosome [49,53,54]. The production of PI(3,4)P$_2$ on early endosome subdomains has been proposed to play a role in recruiting and activating Rab11 and directing the transport of cargo to recycling pathways [55,56]. PI(3,4)P$_2$ has also been implicated in the recruitment of proteins that promote vesicle remodeling and scission [6,29,38,48,57–59]. However, the principle means by which PI(3,4)P$_2$ is produced at these membranes is unclear, although there is evidence that PI(3,4)P$_2$ can be synthesized directly by class II PI3Ks and is not derived solely from the dephosphorylation of PI(3,4,5)P$_3$ [26,27,29,31,57,58,60,61].

Some challenges inherent to studying the roles of PI3Ks and other PI/PPIns modifiers include the transient nature of PPIns species, the interdependency of pathway modifiers and substrates, and the ability to generate identical PPIns through more than one route [62–64]. Such features can complicate the interpretation of genetic perturbations or the use of selective inhibitors, and highly selective inhibitors are not available for all enzymes including class II PI3Ks. [18,32,65]. Another challenge is posed by the heterogeneous and dynamic nature the endosomal compartments marked by PPIns, which can consist of multiple subpopulations and often contain differentially marked subdomains on their cytosolic surfaces [48,66,67]. Finally, the ability to visualize PPIns inside cells using antibodies has not been widely adopted because the fixation methods required preclude live imaging, which is invaluable for determining the localization, dynamics, and functions of these small lipids [62,68]. This last challenge has been addressed by the development of genetically encoded

lipid biosensors, which tether fluorescent proteins to lipid-binding domains that can recognize individual PPIns species [69–71]. Nevertheless, the use of lipid biosensors is complicated by their cross-specificity with different lipids, by difficulties with matching sensor levels to the abundance of the PPIns species, and by adverse physiological effects caused by the sensors, which may compete with native PPIns-binding proteins for limited sites [62,69–71]. Despite these complications, lipid biosensors are currently the best tool available for visualizing intracellular PPIns pools and understanding their roles in membrane trafficking.

In this study we use the nematode, *Caenorhabditis elegans*, as an established model for studying membrane trafficking within a live intact organism [72]. Membrane trafficking is closely coupled to the *C. elegans* molting cycle, as the inhibition of trafficking proteins can lead to strong defects in the molting process [73]. Previously, we have shown that two conserved NIMA-related Ser/Thr protein kinases, NEKL-2 (NEK8/9 in mammals) and NEKL-3 (NEK6/7), and their binding partners, the conserved ankyrin repeat proteins MLT-2, MLT-3, and MLT-4 (mammalian ANKS6, ANKS3, and INVS, respectively), are essential for proper molting, in large part through their roles in controlling membrane trafficking [74–78]. We also described a genetic approach for identifying suppressors of *nekl*-associated molting defects, which led to the identification of several conserved regulators of membrane trafficking, including TAT-1 (mammalian ATP8A1/2), a phosphatidylserine flippase, along with components and regulators of clathrin mediated endocytosis [75,79].

Here we report the identification of mutations affecting PIKI-1, a *C. elegans* ortholog of mammalian class II PI3Ks, which comprise three family members (PI3KC2A, PI3KC2B, PI3KC2G) [10,12,16,21]. Importantly, PIKI-1 is the sole predicted class II PI3K in *C. elegans*, allowing us to study class II PI3K functions in the absence of genetic redundancy caused by paralogs. We found that reduction of PIKI-1 activity led to defects in several endocytic compartments, most notably the early endosome. To better understand the cellular functions of PIKI-1, we generated PPIns biosensors to visualize lipid pools in the epidermis of adult worms. Our study indicated that whereas PIKI-1 is a minor contributor to PI(3)P pools in the epidermis, it is a major contributor to $PI(3,4)P_2$ levels, supporting the model that $PI(3,4)P_2$ is predominantly synthesized by class II PI3Ks and not through the degradation of class I PI3K–derived products. We also provide evidence that loss of *piki-1* can alleviate defects caused by loss of NEKL-2 function at early endosomes, suggesting a mechanism by which *piki-1* mutations may suppress *nekl*–associated molting defects.

## Results

### *nekl* reduction-of-function molting defects are suppressed by mutations in *piki-1*

We previously described a forward genetic screen and whole-genome sequencing pipeline to identify genetic suppressors of larval lethality caused by *nekl* reduction-of-function mutations [80]. Briefly, this approach exploits a synthetic lethal interaction that occurs when two aphenotypic (weak) reduction-of-function alleles of *nekl-2(fd81)* and *nekl-3(gk894345)* are combined in the same animal. *nekl-2(fd81)*; *nekl-3(gk894345)* (hereafter *nekl-2; nekl-3*) worms showed highly penetrant molting defects, with ~98% of progeny arresting at the L2/L3 boundary and only ~2% progressing to adulthood (Fig 1D and 1F). *nekl-2; nekl-3* worms can be propagated in the presence of a rescuing extrachromosomal array (*fdEx286*) containing wild-type copies of *nekl-3* and a *sur-5::GFP* reporter to facilitate the visualization of array-positive animals (Fig 1C). Worms failing to inherent the array in the parental strain exhibited a "corset" phenotype, whereby old cuticle failed to be consistently shed from midbody (Fig 1D). Following mutagenesis, worms containing suppressor mutations were identified by their ability to reach adulthood and propagate robustly in the absence of the rescuing array (Fig 1E).

From our suppressor screen, we identified three independent alleles affecting the *piki-1* locus (*fd161*, *fd275*, and *fd357*), such that ~50–65% of *nekl-2; nekl-3 piki-1* worms reached adulthood (Fig 1F). Two alleles, *fd161* and *fd357*, both cause a C-to-T transition in the second-to-last exon of *piki-1*, resulting in premature stop codons at amino acids 1507 and 1503, respectively (Fig 1G and 1H). The third allele, *fd275*, is a G-to-A transition in the fifth exon of *piki-1*, leading to the substitution of asparagine for serine at amino acid 211 (Fig 1G and 1H). *fd161* and *fd357* would be expected to remove the terminal C2B domain, which binds $PI(4,5)P_2$ and $PI(3,4,5)P_3$, facilitating membrane localization and likely impacting

enzymatic function [81]. *fd161* and *fd357* may also be subject nonsense-mediated decay, resulting in reduced mRNA and protein levels in addition to truncation of the protein [82]. S211 is in a non-conserved region of PIKI-1 that is predicted to be largely unstructured; its effects on PIKI-1 function are currently unknown.

CRISPR phenocopy of *fd161* (*fd381*) led to levels of suppression that were similar to those observed for *fd161* (Fig 1F), demonstrating that *piki-1* is the causal locus in the *nekl-2; nekl-3 piki-1(fd161)* strain. Additionally, *fd161* failed to complement *fd275*, consistent with *piki-1* being the causal locus in both strains. Moreover, a consortium-generated 1597-bp deletion mutation [83] in *piki-1* (*ok2346*) led to 66% of *nekl-2; nekl-3 piki-1(ok2346)* worms reaching adulthood (Fig 1F–1H). Using CRISPR, we introduced a mutation into the catalytic domain of PIKI-1 (D1189A) that is expected to strongly reduce PIKI-1 kinase activity [13] and observed ~50% viability of *nekl-2; nekl-3* adults, indicating that PIKI-1 enzymatic activity is critical to its function and relevant to its genetic interactions with the *nekls*. Conversely, we failed to observe robust suppression of *nekl-2; nekl-3* defects by *piki-1(RNAi)* using dsRNA injection methods, which is likely due to insufficient knockdown of *piki-1* (4.8% viability; S1A Fig). Finally, *piki-1*(Q1507Stop) failed to suppress a stronger loss-of-function *nekl-2* allele, *fd91* (0% viability; n = 139), indicating that *piki-1* mutations may suppress only a subset of partial reduction-of-function mutations in the *nekls*.

PIKI-1 belongs to a family of lipid kinases that specifically phosphorylate the 3' hydroxyl position on the inositol ring of membrane PIs and PPIns (Fig 1A) [12,16]. More specifically, PIKI-1 is the sole member of the class II PI3Ks in *C. elegans*, which in other species have been reported to convert PI to PI(3)P and PI(4)P to $PI(3,4)P_2$ (Fig 1B) [19–21]. Given the proposed role of PIKI-1 in PI/PPIns modification, we tested several additional (non-essential) PPIns modifiers for genetic interactions with the *nekls*. Given that $PI(3,4)P_2$ can be derived from $PI(3,4,5)P_3$ after removal of the 5' phosphate (Fig 1B), we tested AGE-1, the sole class I PI3K in *C. elegans* and producer of $PI(3,4,5)P_3$ [84,85]. We found that inhibition of *age-1* by RNAi (using dsRNA injection methods) or a reduction-of-function mutation (*hx546*) had little or no ability to suppress molting defects in *nekl-2; nekl-3* mutants (S1A and S1B Fig). Likewise, RNAi of *inpp-1*, a putative 5' phosphatase predicted to convert $PI(3,4,5)P_3$ to $PI(3,4)P_2$, also failed to promote strong suppression (7.8% viability; S1A Fig). Nevertheless, RNAi of *inpp-1* led to a ~45% reduction in brood size in the F1 progeny of injected *nekl-2; nekl-3* worms (S1 File), indicating that inhibition of INPP-1 by RNAi methods impacts fertility. Collectively, our results suggest that loss of *piki-1* may be unique in its ability to robustly suppress *nekl* defects, although our inability to test essential PPIns modifiers, along with caveats associated with partial knockdown by RNAi, limit these conclusions.

## PIKI-1 localizes to clathrin-coated pits and early endosomes

Previously, we reported colocalization of NEKL-2 and NEKL-3 with several internal membrane-bound compartments, consistent with roles for the NEKLs in endocytic trafficking [74]. More specifically, NEKL-2 localizes most extensively to early endosomes, whereas NEKL-3 resides predominantly at late endosomes. Mammalian class II PI3Ks localize to early endosomal compartments and to clathrin-coated structures, as well as to recycling endosomes [21,86,87]. In *C. elegans*, PIKI-1 localizes to nascent phagosomes in embryos and in the germline [88–92].

To characterize the endogenous localization of PIKI-1 in the epidermis, we first examined CRISPR-generated fusions of either GFP or mScarlet to the C terminus of PIKI-1. However, endogenous PIKI-1::GFP proved too dim for reliable imaging (S2A Fig) and PIKI-1::mScarlet showed non-specific localization to lysosomes (S2B Fig), which is likely due to cleavage of the fluorophore and its retention in lysosomes [93]. We note, that PIKI-1::GFP retains at least partial activity as we failed to observe suppression of molting defects in *nekl-2; nekl-3 piki-1::GFP* worms (0% viability; n = 226). To enable visualization of PIKI-1, we integrated a single-copy PIKI-1::mNeonGreen transgene driven by a promoter expressed predominantly in the hyp7 epidermal syncytium ($P_{hyp7}$; *semo-1*) using miniMos methods [94]; *semo-1* is expressed at ~10-fold higher levels than *piki-1* in young adults [95,96]. $P_{hyp7}$::PIKI-1::mNeonGreen was expressed in the adult epidermis both diffusely and in punctate structures located near the apical plasma membrane (Fig 2A, 2B, 2B', 2D and 2D') as well as in more irregularly shaped structures located apically but below the membrane (Fig 2I and 2I'). Based on the distribution and

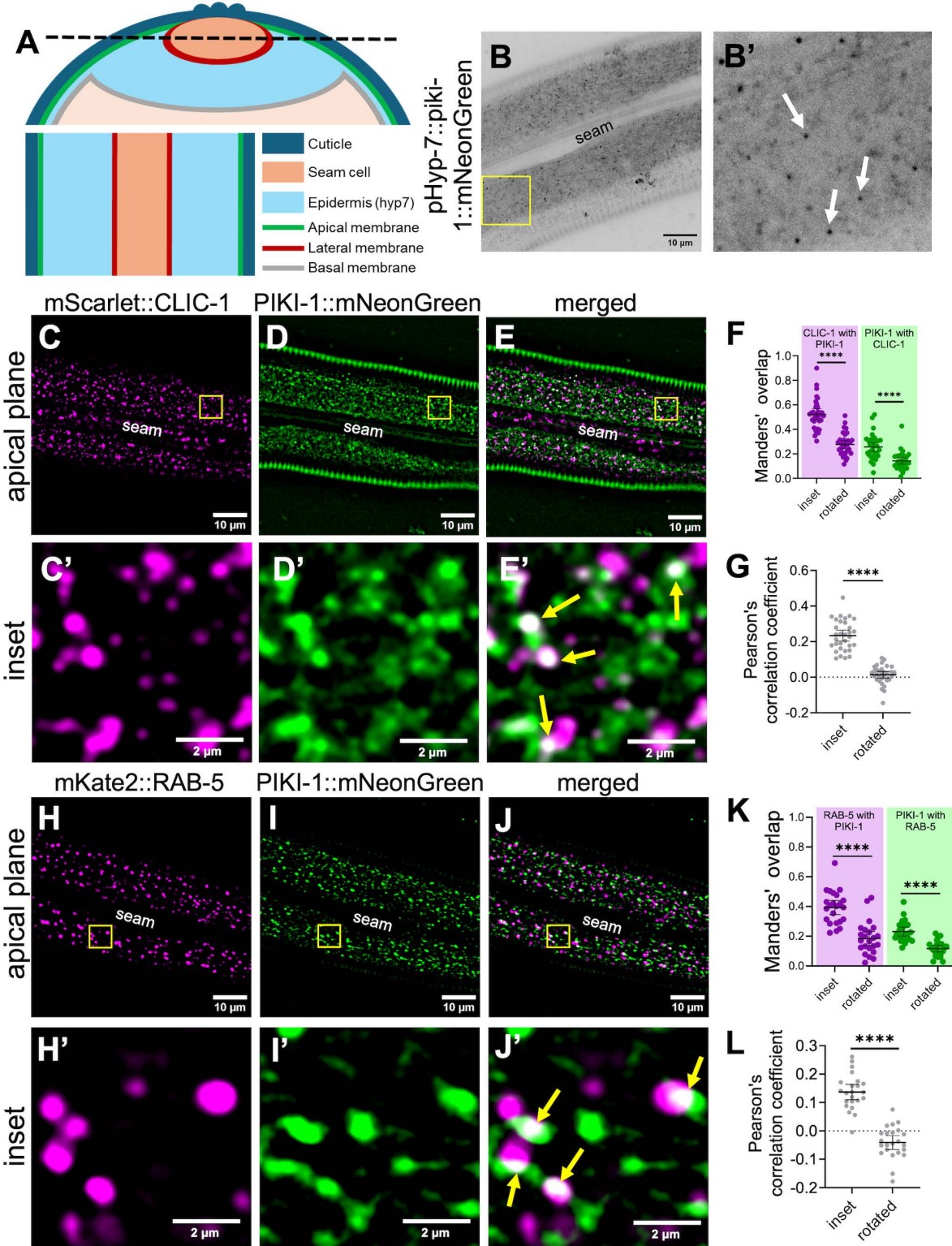

**Fig 2. PIKI-1 colocalizes with clathrin-coated pits and the early endosome.** (A) Schematic diagram of an adult *C. elegans* as a transverse cross-section (top) and a longitudinal cross-section (bottom), which corresponds to figure panels. The black dashed line on the transverse cross-section denotes the apical plane visualized in the longitudinal cross-section for a single z-slice. Bottom right contains a key indicating the relevant cell types and

membranes. (B, B') Representative unprocessed images of a day-1 adult expressing $P_{hyp7}$::PIKI-1::mNeonGreen. White arrows indicate apical puncta. (C–E') Colocalization of day-1 adult worms expressing (C, C') CRISPR CLIC-1::mScarlet and (D, D') $P_{hyp7}$::PIKI-1::mNeonGreen (n = 32); (E, E') merged images. (H–J') Trans-heterozygous worms expressing (H, H') $P_{dyp-7}$::mKate2::RAB-5 and (I, I') $P_{hyp7}$::PIKI-1::mNeonGreen (n = 23); (J and J') merged images. The seam cell is labeled in lower-magnification images. Yellow squares (C–E and H–J) indicate the locations of the enlarged insets (C'–E' and H'–J'). In merged insets (E', J'), yellow arrows indicate examples of colocalization (white). (F, G, K and **L**) Colocalization was quantified using Mander's overlap (F and K) and Pearson's correlation coefficients **(G, L)**. Dot plots show the mean and 95% CI. Statistical significance between rotated and inset values was determined using unpaired *t*-tests; ****p ≤ 0.0001. Raw data are available in S1 File.

size of $P_{hyp7}$::PIKI-1::mNeonGreen puncta, we hypothesized that PIKI-1 may localize to clathrin-coated pits–vesicles and to apical endosomes, consistent with prior studies on mammalian class II PI3Ks.

As anticipated, we observed colocalization between a marker for clathrin light chain, mScarlet::CLIC-1, and PIKI-1::mNeonGreen (Fig 2C–2E'). However, whereas ~50% of the mScarlet::CLIC-1 signal overlapped with the PIKI-1::mNeonGreen signal (Manders' overlap = ~0.5), only ~25% of PIKI-1::mNeonGreen overlapped with mScarlet::CLIC-1 (Manders' overlap = ~0.25) (Fig 2F), indicating that PIKI-1 localizes to additional membrane compartments. Significant positive but partial overlap between the markers was also supported by the Pearson's correlation coefficient (PCC = ~0.23) (Fig 2G). As an additional control for all colocalization experiments, we rotated one of the two channels and reanalyzed the Manders' and Pearson's values and observed a dramatic reduction in both measurements, indicating that the observed overlap was non-random (Fig 2F and 2G).

We next examined colocalization between PIKI-1::mNeonGreen and a marker for early endosomes, $P_{dyp-7}$::mKate2::RAB-5. In animals that were heterozygous for both markers (also see Materials and Methods) we found that ~40% of the mKate2::RAB-5 signal overlapped with PIKI-1::mNeonGreen (Manders' overlap = ~0.4; Fig 2H–2K), whereas ~23% of the PIKI-1::mNeonGreen signal overlapped with mKate2::RAB-5 (Manders' overlap = ~0.23; Fig 2H–2K). Additionally, the Pearson's correlation coefficient (PCC = ~0.14) is consistent with partial overlap between mKate2::RAB-5 and PIKI-1::mNeonGreen (Fig 2L). These results indicate that PIKI-1 is associated with both clathrin-coated pits and endosomal trafficking compartments, raising the possibility that class II PI3K functions in endomembrane trafficking may be partially conserved between *C. elegans* and mammals. Additionally, $P_{hyp7}$::PIKI-1::mNeonGreen likely localizes to additional endomembrane or other cytoplasmic compartments.

We next tested PIKI-1::mNeonGreen colocalization with four additional endomembrane compartment markers including GFP::RAB-7 (late endosomes), NUC-1::mCherry (lysosomes), $P_{hyp7}$::mScarlet::RME-1 (RME-1–positive recycling endosomes), and CHAT-1::mKate (early endosomes and CHAT-1–positive recycling endosomes) [79,97–99]. Notably, we did not observe strong co-localization with any of these markers, although PIKI-1 overlapped weakly with RAB-7 (S3 Fig) and CHAT-1 puncta often appeared adjacent to PIKI-1 puncta (S4 Fig). The slight overlap between PIKI-1 and RAB-7 could be explained by the partial localization of RAB-7 to "early" endosomes during the endosomal conversion process (early-to-late stages) [100–103]. Moreover, flanking PIKI-1 and CHAT-1 signals may reflect the localization of these proteins to distinct sub-domains on early endosomes or to emergent recycling endosomes labeled by CHAT-1. Collectively, our findings indicate PIKI-1 localizes primarily to early endosomes and clathrin-coated pits but does not rule out targeting to additional endomembrane compartments.

## Loss of PIKI-1 impacts a subset of endosomal compartments

Mammalian class II PI3Ks and the production of PI(3,4)$P_2$ are implicated in the scission of nascent clathrin-coated vesicles and the subsequent uncoating and maturation of vesicles enroute to the early endosome [41,86,104]. To determine whether PIKI-1 regulates early steps of clathrin-mediated endocytosis in *C. elegans*, we examined markers for clathrin light and heavy chains in wild type and *piki-1* mutants. Despite localization of PIKI-1 to clathrin-coated structures, *piki-1(Q1507Stop)* mutants showed a wild-type-like pattern of localization for both clathrin heavy (GFP::CHC-1) and light

(mScarlet::CLIC-1) chains, suggesting that PIKI-1 does not play a major role in clathrin-mediated endocytosis in the worm epidermis (S5A–S5F Fig). Consistent with this, LRP-1, an apically expressed low-density lipoprotein–like receptor that is trafficked through apical clathrin-coated pits [105], showed only slightly increased accumulation at the apical surface in *piki-1(Q1507Stop)* worms as compared with wild type (S5G–S5I Fig). Together these data indicate that class II PI3K activity may be largely dispensable for clathrin-mediated endocytosis in the *C. elegans* epidermis.

After internalization at the plasma membrane, cargoes are sorted at early endosomes for subsequent routing to either recycling or degradative pathways (Fig 3A) [48,72]. To investigate the role of PIKI-1 at early endosomes, we examined two early endosomal markers, GFP::RAB-5 and GFP::EEA-1, in wild type and *piki-1* mutants. RAB-5 is a small GTPase that recruits proteins required for endocytic sorting and transport, whereas EEA-1 is a conserved effector of RAB-5 that promotes vesicle docking and fusion (Fig 3A) [51,52]. Notably, we observed a marked (~1.5-fold) decrease in the number of RAB-5– and EEA-1–positive endosomes in *piki-1(Q1507Stop)* mutants as compared with wild type (Fig 3B–3D and 3F–3H). Moreover, the size of RAB-5– and EEA-1–positive puncta was reduced by ~1.5- and ~1.3-fold, respectively (Fig 3E and 3I). In contrast, reduction of PIKI-1 function had no detectable effect on the mean intensity or morphology of late endosomes marked by GFP::RAB-7 (S6A–S6D Fig), consistent with localization of PIKI-1 to early but not late endosomes (S3 Fig). These findings suggest that PIKI-1 plays a role in the biogenesis, homeostasis, or organization of early endosomes.

To further investigate the role of PIKI-1 at early endosomes we examined the localization of the sorting nexin SNX-1 in the epidermis. SNX-1 is an ortholog of human SNX1 and SNX2, which bind PI(3)P and promote membrane bending and tubulation in part through interactions with the retromer complex, facilitating tubulation, cargo sorting, and the recycling of cargo from early endosomes to the trans-Golgi and plasma membrane [106–110]. In *C. elegans* SNX-1 has been shown to localize to a sub-domain/population of early endosomes in intestinal cells and coelomocytes, consistent with localization data for SNX1 and SNX2 in mammalian cells [107,111–114]. Likewise, we observed partial co-localization of a P$_{hyp7}$:: mScarlet::SNX-1 epidermal reporter with both GFP::RAB-5 and PIKI-1::mNeonGreen (Fig 4). In contrast to RAB-5 and EEA-1, however, loss of *piki-1* led to a ~1.4-fold increase in the average mean intensity of a P$_{hyp7}$::mNeonGreen::SNX-1 epidermal reporter and to a ~1.2-fold increase in the density of SNX-1–marked puncta (Fig 3J–3M). Taken together, our data suggest that PIKI-1 promotes RAB-5 and EEA-1 association while restricting SNX-1 binding, thereby influencing the composition of membrane domains at early endosomes.

Several studies have implicated mammalian Class II PI3Ks in regulating cargo recycling pathways [55,87,115]. To assess the role of PIKI-1 in *C. elegans* endocytic recycling, we examined the localization of GFP::RAB-11 and mNeonGreen::RME-1 in *piki-1* mutants. In the worm epidermis, RAB-11 marks a basal–medial recycling compartment, which may be equivalent to the endocytic recycling compartment (ERC) in mammals, whereas RME-1 marks a distinct population of apical recycling endosomes [74,79]. Notably, reduction of PIKI-1 had no obvious effects on RME-1-marked endosomes and only minimal effects on the GFP::RAB-11 marker including a ~1.1-fold decrease in mean intensity (p = 0.0012) and a modest (though not statistically significant) decrease (~1.3-fold; p = 0.15) in the size of the GFP::RAB-11 puncta (S6E–S6L Fig). As such, PIKI-1 appears to have at most a minor or indirect role on the assayed recycling compartments.

We also examined localization of the cargo protein TGN-38, which travels from the trans Golgi to the plasma membrane during exocytosis and is recycled back to the Golgi via early endosomes, potentially bypassing the RAB-11 compartment [116,117]. Although no changes in mean intensity of TGN-38::GFP were observed between wild type and *piki-1(Q1057Stop)* mutants (S5J Fig), we noticed a striking tubulation phenotype, as marked by TGN-38::GFP, in ~50% of *piki-1(Q1507Stop)* worms (Fig 3N–3P). Additionally, we detected a ~30% decrease in the percentage of TGN-38::GFP– positive pixels (above threshold) in *piki-1(Q1057Stop)* animals (Fig 3Q), suggesting a redistribution and accumulation of TGN-38 to more tubular structures. These data suggest that PIKI-1 plays a role in the transit of TGN-38 through the endocytic sorting and recycling pathways, although the compartmental identity of the TGN-38–positive tubules is currently unknown.

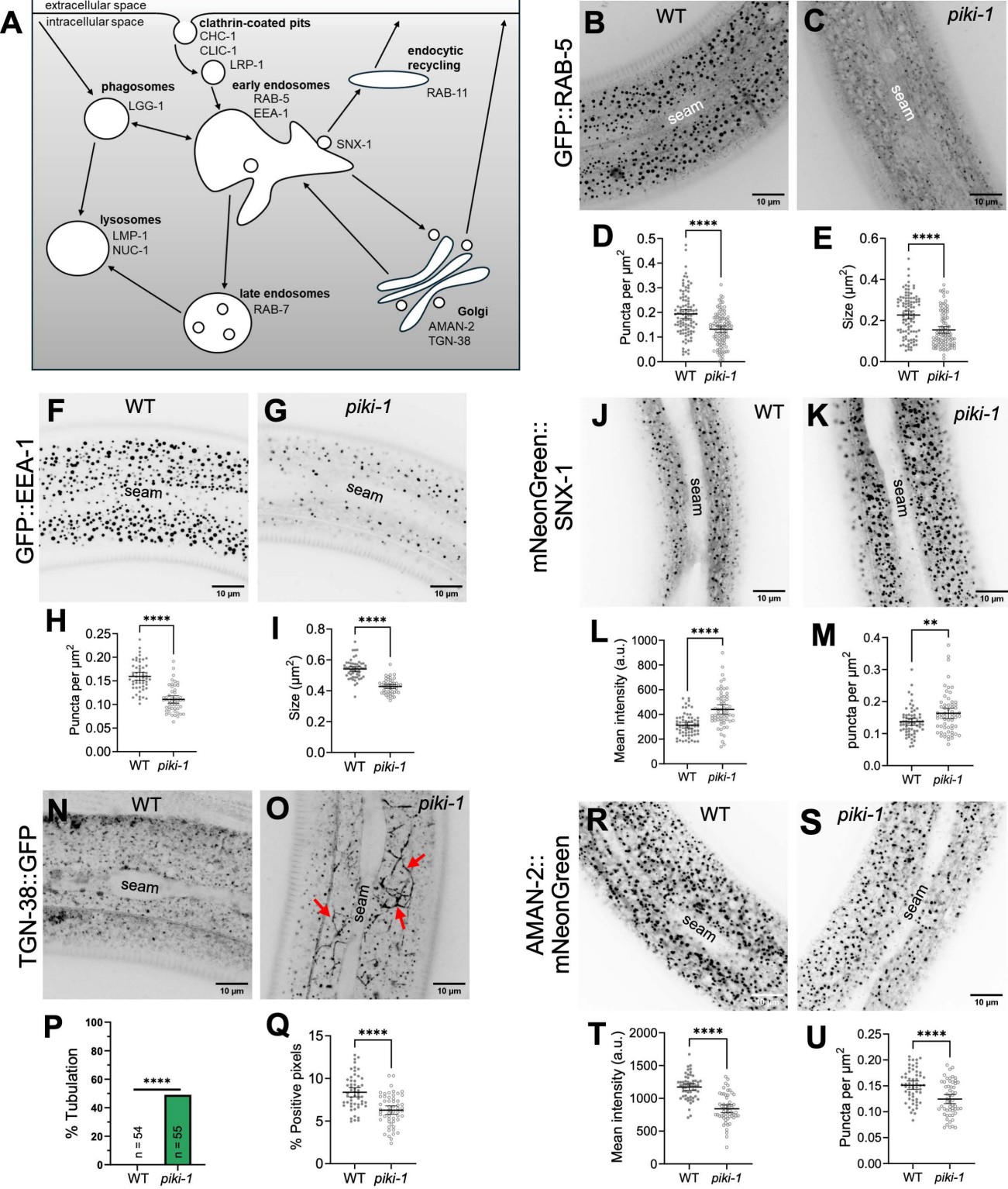

**Fig 3. Reduction of PIKI-1 function affects different endosomal compartments. (A)** Schematic of endocytosis and associated endosomal compartments. Compartments are labeled in bold with associated proteins and cargos indicated. (B, C,F, G, H, J, K, N, O, R, and **S)** Representative confocal images of day-1 adults that were used to assess the markers **(B, C)** P$_{rab-5}$::GFP::RAB-5, **(F, G)** P$_{hyp7}$::GFP::EEA-1, **(J, K)** P$_{hyp7}$::mNeonGreen::SNX-1, **(N,**

O) P$_{hyp7}$::TGN-38::GFP, and (R, S) P$_{hyp7}$::AMAN-2::mNeonGreen in wild-type and *piki-1(Q1507Stop)* mutants. The seam cell is labeled in all images. Calculated metrics included (D, H, M and U) number of puncta per unit area; (E, I) Size (area) of puncta; (L, T) average mean intensity; (Q) percentage of positive pixels (above threshold). All dot plots show the mean and 95% CI. Statistical significance was determined using unpaired *t*-tests; ****p ≤ 0.0001; **p ≤ 0.01. (O) Red arrows indicate tubulations marked by TGN-38::GFP. (P) Statistical significance for tubulations was determined using Fisher's exact test; ****p ≤ 0.0001. Raw data are available in S1 File.

Given our findings for TGN-38, we assessed the impact of PIKI-1 loss on the Golgi using a AMAN-2::mNeonGreen marker [118]. AMAN-2 is a homolog of mammalian alpha-mannosidase II (Man II), a resident Golgi protein that has been shown to be distributed throughout the Golgi in a cell type–dependent fashion [119,120]. Notably, we observed an ~1.4-fold reduction in the mean intensity of AMAN-2::mNeonGreen in *piki-1(Q1507Stop)* mutants along with a modest decrease in the number of AMAN-2 puncta (~1.2-fold) (Fig 3R–3U) but with relatively little change in morphology. We note that although our available markers did not allow us to directly test for PIKI-1 localization to Golgi, the distinct locations of AMAN-2::mNeonGreen and PIKI-1::mNeonGreen puncta along the apicobasal axis of hyp7 was not suggestive of a substantial overlap. Our results suggest that defects in the sorting and/or transport of cargo from the early endosome to the Golgi in *piki-1* mutants may indirectly impact the size of the Golgi compartment.

PIKI-1 has been reported to promote the clearance of phagosomes in the developing embryo and adult germline through the production of PI(3)P [88,90–92]. We therefore asked whether PIKI-1 was required for the normal localization of the downstream phagosome/autophagy effector, LGG-1, in the adult hyp7 (Fig 3A) [121–124]. We observed a modest reduction (~1.3-fold) in the mean intensity of mNeonGreen::LGG-1 in *piki-1(Q1507Stop)* mutants relative to wild type as well as an increase in the frequency of elongated or tubular structures marked by LGG-1 (38%) versus wild type (10%) (S6M–S6P Fig). Although suggestive of a function for PIKI-1 in autophagy, the nature of the compartment marked by mNeonGreen::LGG-1 is currently unclear. Collectively, our data suggest several roles for PIKI-1 in membrane trafficking in hyp7 including the maintenance of early endosome subdomains, cargo recycling from the early endosome to the Golgi, and a potential role in canonical or non-canonical autophagy.

## Loss of PIKI-1 suppresses defects in early endosomes associated with NEKL-2 depletion

Based on the above findings, we were curious as to how loss of *piki-1* might lead to the genetic suppression of *nekl-2; nekl-3* mutants. We had previously observed localization of NEKL-2 to early endosomes (GFP::RAB-5) and shown that depletion of NEKL-2, using the auxin-inducible degron (AID) system, results in an expansion of the RAB-5–marked early endosome compartment [74]. To determine if loss of *piki-1* alleviates defects at early endosomes following NEKL-2::AID depletion, we compared GFP::RAB-5 localization in auxin-treated *nekl-2::AID* and *nekl-2::AID; piki-1(Q1507Stop)* adults. Consistent with prior findings, we observed ~1.4-fold increases in both total mean intensity and puncta size as well as a ~1.2-fold increase in the number of GFP::RAB-5 puncta following NEKL-2::AID depletion (Fig 5A, 5B, and 5D–5F). Notably, this expansion of the GFP::RAB-5 compartment was prevented when NEKL-2::AID was depleted in the *piki-1(Q1507Stop)* background. Specifically, we observed a decrease in the mean intensity (~1.3-fold), size (~1.8-fold), and number of RAB-5–marked puncta (~1.2-fold) in *nekl-2::AID; piki-1(Q1507Stop)* adults relative to *nekl-2::AID* alone (Fig 5B–5F). Overall, the early endosome compartment in NEKL-2–depleted *piki-1(Q1507Stop)* worms was comparable to wild-type, albeit with a modest reduction in puncta size (~1.3-fold). The reversal of *nekl-2*-associated early endosome defects by *piki-1* suggests a possible means by which *piki-1* mutations may suppress *nekl-2; nekl-3* molting defects.

In contrast, loss of *piki-1* did not alleviate clathrin localization defects following depletion of AID-tagged NEKL-3. Consistent with our prior work [75] we observed an increase in apical GFP::CHC-1 following NEKL-3::AID depletion, leading to an a ~1.3-fold increase in mean intensity and a 2.4-fold increase in the percentage of CHC-1::GFP–positive pixels (above threshold) (Fig 5G, 5H, 5J, and 5K). Similarly, NEKL-3::AID–depleted *piki-1(Q1507Stop)* worms showed a ~1.4-fold

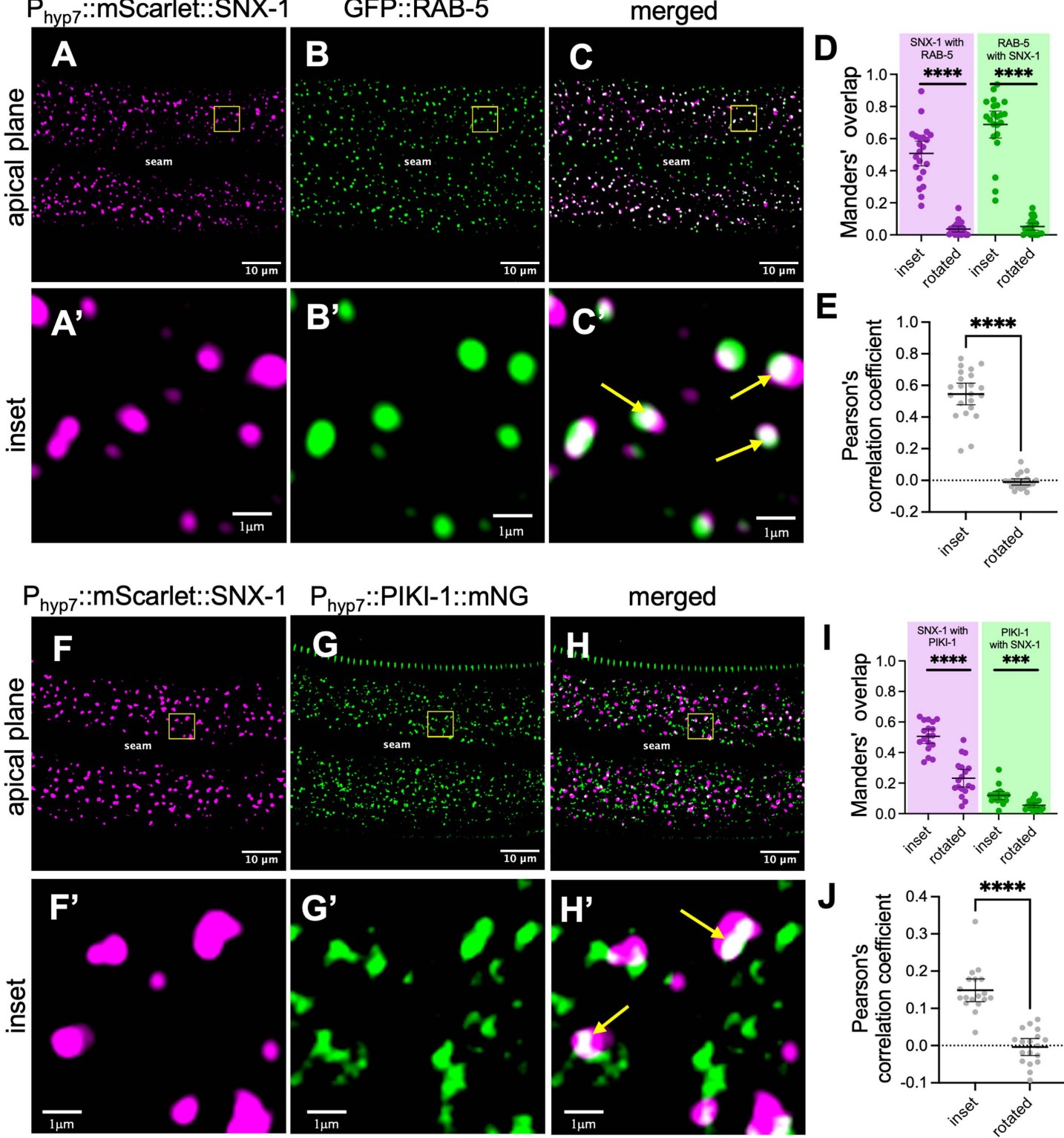

**Fig 4. SNX-1 colocalizes with the early endosome markers RAB-5 and PIKI-1.** (A–C' and F–H') Colocalization of young adult worms expressing heterozygous $P_{hyp7}$::mScarlet::SNX-1 with (A–C') heterozygous GFP::RAB-5 (n = 22) or (F–H') heterozygous $P_{hyp7}$::PIKI-1::mNeonGreen (n = 18). Yellow

squares correspond to enlarged insets; yellow arrows indicate examples of colocalization (white). The seam cell is labeled in A–C and F–H. Colocalization was quantified using Mander's overlap (D, I) and Pearson's correlation coefficients **(E, J)**. Dot plots show the mean and 95% CI. Statistical significance between rotated and inset values was determined using unpaired *t*-tests; ****p ≤ 0.0001; ***p ≤ 0.001. Raw data are available in S1 File.

increase in mean intensity and a ~2.2-fold increase in the percentage of positive pixels (Fig 5G, 5I, 5J, and 5K). Taken together, our results indicate that loss of PIKI-1 may alleviate a subset *nekl*-associated trafficking defects, most notably those associated with early endosomes.

## Loss of PIKI-1 alters polyphosphoinositide pools in the epidermis

To understand more precisely how PIKI-1 regulates membrane phospholipids, we generated lipid sensors to investigate the distribution of PPIns species in wild type and *piki-1* mutants. Class II PI3Ks contribute to the generation of two distinct lipid species, PI(3)P and PI(3,4)$P_2$, which in turn recruit specific proteins to cytosolic-facing membranes [21]. We expressed fluorescently tagged lipid biosensors under the control of epidermal-specific promoters ($P_{hyp7}$ or $P_{nekl-3}$) [79,125,126], where each biosensor was integrated as a single copy using miniMos methods or was maintained as an extrachromosomal array (see S2 File and Materials and Methods) [94].

In mammalian cells, PI(3)P is associated with early endosomes as well as with autophagosomes and nuclei [127]. Consistent with this, our PI(3)P sensor ($P_{hyp7}$::2xFYVE::mNeonGreen) localized to punctate structures close to the apical surface in wild-type adults as well as to more basally located hyp7 nuclei (Fig 6A). *piki-1(Q1507Stop)* mutants exhibited a similar pattern of PI(3)P localization, although we detected a slight decrease (~1.2-fold) in the number of vesicles relative to wild type (Fig 6B and 6C). Likewise, the *piki-1(ok2346)* deletion mutant displayed a similar modest decrease in the number of PI(3)P-marked vesicles (S7B–S7D Fig). The absence of a strong effect on PI(3)P is consistent with previous work demonstrating that class III PI3Ks can be the predominant producers of PI(3)P [10,92]. To further examine PI(3)P localization, we carried out colocalization analysis of the PI(3)P sensor with an early endosome marker, mKate2::RAB-5 [128]. Approximately 32% of the mKate2::RAB-5 signal overlapped with our PI(3)P sensor (Manders' overlap = 0.32) and ~65% of PI(3)P-labeled puncta overlapped with the mKate2:RAB-5 signal (Manders' overlap = 0.65; Fig 7A–7D). This substantial overlap is further supported by the Pearson's correlation coefficient (PCC = 0.37; Fig 7E). Thus, PI(3)P pools, which were slightly reduced by loss of PIKI-1, are associated with early endosomes, consistent with our findings for PIKI-1 localization. Our findings also suggest that PI(3) localizes to membrane domains or compartments other than RAB-5–marked endosomes and that portions of RAB-5–positive endosomes may not contain substantial levels of PI(3).

Like PI(3)P, PI(3,4)$P_2$ has been associated with early endosomes as well as with endocytic recycling compartments [23,56]. Additionally, PI(3,4)$P_2$ is found on the apical membrane of polarized epithelial cells [58]. In wild type, the PI(3,4)$P_2$ sensor ($P_{nekl-3}$::2xTAPP1::mNeonGreen; *fdSi8*) exhibited a diffuse localization pattern in both apical and basal planes but also accumulated at apical puncta (Fig 6D and 6D' and S1 Movie). We note that the diffuse expression of the PI(3,4)$P_2$ sensor may be due in part to the sensor being present in excess of its preferred ligand, a known caveat associated with genetically encoded lipid sensors [26]. Strikingly, reduction of PIKI-1 function led to a 1.9-fold decrease in the amount of the PI(3,4)$P_2$ present in the epidermis (Fig 6D–6F), consistent with PIKI-1 acting as a major producer of PI(3,4)$P_2$. The decreased abundance of the lipid-sensor may be attributable to a reduction in lipid-binding sites and increased reporter turnover, as has been reported [69–71,129,130]. Correspondingly, there was no observed accumulation of the lipid sensor at a more basal plane in *piki-1(Q1507Stop)* mutants (S2 Movie).

Notably, our PI(3,4)$P_2$ sensor (*fdEx406*) showed substantial colocalization with the mKate2::RAB-5 marker; ~62% of the mKate2::RAB-5 signal overlapped with the PI(3,4)$P_2$ sensor and ~25% of the PI(3,4)$P_2$ sensor overlapped with mKate2::RAB-5 (Fig 7F–7I). Significant positive but partial overlap was also supported by the Pearson's correlation coefficient (PCC = 0.16; Fig 7J). These results suggest that a sizeable proportion of early endosomes are marked by

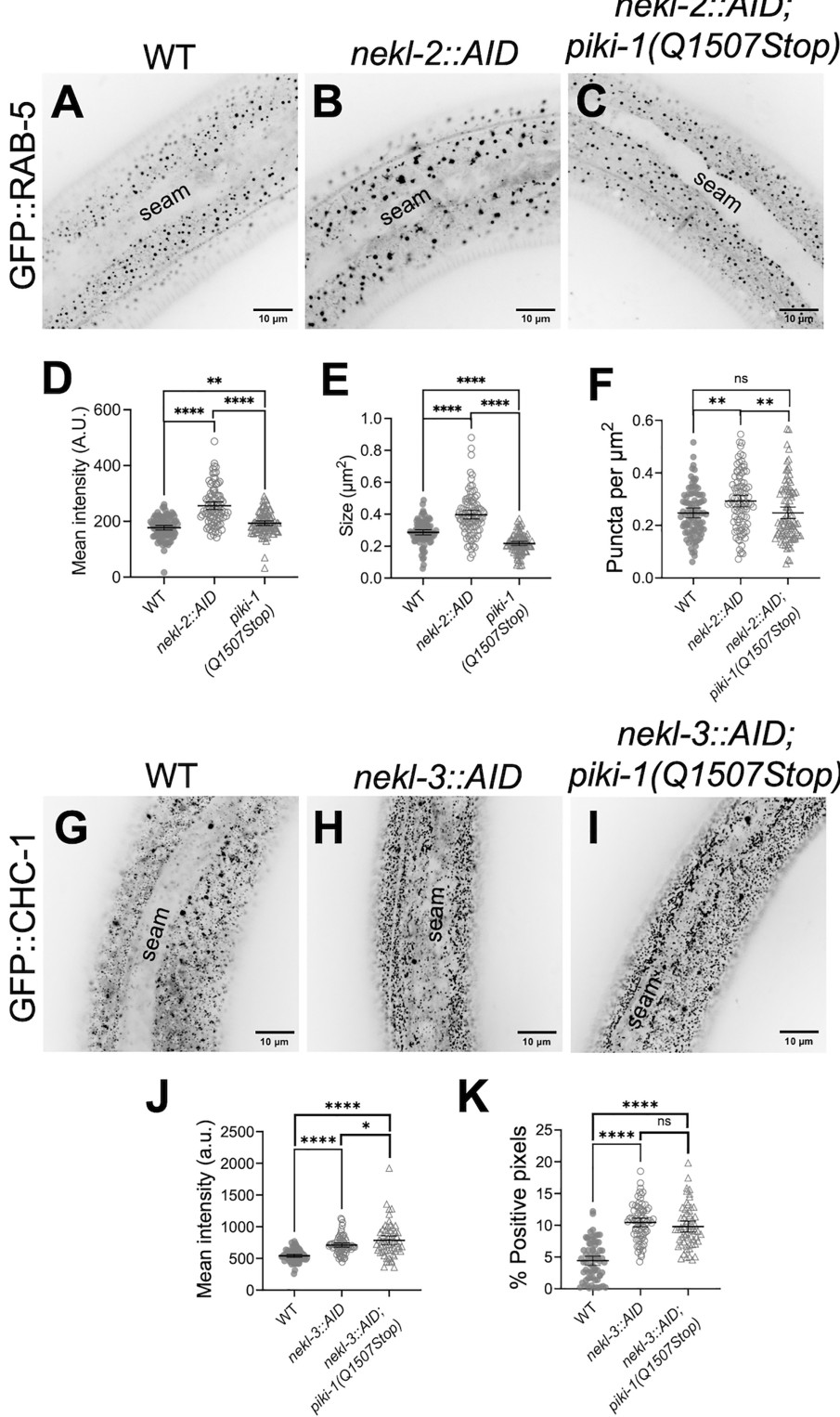

**Fig 5. Loss of PIKI-1 suppresses *nekl-2*–associated defects in the early endosome. (A–C)** Representative confocal images of P*rab-5*::GFP::RAB-5 expression in auxin-treated day-2 adults in (A) wild-type, (B) *nekl-2::AID*, and (C) *nekl-2::AID; piki-1(Q1507Stop)* backgrounds. **(D–F)** Dot plots show (D) mean intensity, (E) puncta size, and (F) the number of puncta per unit area for P*rab-5*::GFP::RAB-5. **(G–I)** Representative confocal images of GFP::CHC-1

expression in auxin-treated day 2 adults in (G) wild-type, (H) *nekl-3::AID*, and (I) *nekl-3::AID; piki-1(Q1507Stop)* backgrounds. **(J, K)** Dot plots show the (I) mean intensity and (J) percent positive pixels (above threshold). (A–C and G–I) The seam cell is labeled in all images. (D–F and J, **K**) Dot plots show the mean and 95% CI. Statistical significance was determined using unpaired *t*-tests; ****$p \leq 0.0001$, **$p \leq 0.01$, *$p \leq 0.05$; ns, not significant. Raw data are available in S1 File.

PI(3,4)P$_2$, the production of which is dependent in large part on PIKI-1. As with the PI(3)P sensor, our results indicate that PI(3,4)P$_2$ may localize to specific populations or subdomains of early endosomes but also suggest that PI(3,4)P$_2$ is also present on additional compartment membranes, such as small protrusions and tubules emanating from early endosomes or apical recycling endosomes (Figs 7F–7J and S7A).

The PPIns biosynthesis pathway is highly interconnected, with multiple lipid kinases and phosphatases contributing to the production of individual PPIns species (Fig 1B). As such, changes in the levels of one PPIns species have the potential to affect other pools [131]. To investigate whether loss of PIKI-1 affects other PPIns populations, we generated lipid sensors to visualize PI(4,5)P$_2$ (PH PLC δ::mNeonGreen), a PPIns involved in clathrin-mediated endocytosis in mammalian cells, and PI(3,4,5)P$_3$ (P$_{nekl-3}$::BTK PH::mNeonGreen), a PPIns associated predominantly with the plasma membrane and phagosome formation that can be converted to PI(3,4)P$_2$ [2,3,5,6,12,16,43,49,132]. In wild type, the PI(4,5)P$_2$ sensor localized throughout hyp7 to folded plasma membrane subdomain compartments that likely represent meisosomes [133], where PI(4,5)P$_2$ is known to be expressed (Fig 7H). Notably, this marker was unchanged in *piki-1(Q1507Stop)* and *piki-1(ok2346)* worms (Figs 6H–6J and S7E–S7G). Likewise, no obvious changes were observed for the PI(3,4,5)P$_3$ sensor in *piki-1(Q1507Stop)* mutants, which exhibited a diffuse localization pattern but also accumulated at variably sized puncta throughout the epidermis (Fig 6K–6M). Our combined results indicate that reduction of PIKI-1 most strongly affects PI(3,4)P$_2$ pools with a limited impact on other species of PPIns, most notably PI(3)P. We note that, although not assayed, PI(4)P pools in the epidermis may also be expected to increase in *piki-1* mutants and may contribute to the observed trafficking phenotypes [134–138].

Lastly we tested a putative multi-specific lipid sensor reported to recognize both PI(3,4)P$_2$ and PI(3,4,5)P$_3$ (AKT::ox-GFP), but which may bind to other lipids and endogenous proteins [69,139]. In wild-type animals, the localization of the dual-specificity sensor most closely resembled that of the PI(3,4)P$_2$ sensor, with combined diffuse localization accompanied by distributed puncta (S8A Fig). Curiously, the AKT::oxGFP reporter exhibited a dramatic tubulation phenotype in ~50% of *piki-1(Q1507Stop)* and *piki-1(ok2346)* mutants (S8B–S8G Fig), reminiscent of the phenotype of *piki-1* mutants expressing TGN-38::GFP. Given that lysosomes can form elongated tubules in the epidermis during normal molting cycles [98], we assessed two lysosomal markers, LMP-1::mNeonGreen [127] and NUC-1::mCherry [140], in *piki-1* mutants. LMP-1 is a conserved lysosomal membrane protein involved in lysosome biogenesis [141] whereas NUC-1 is a lysosomal hydrolase that acts within the lysosomal compartment [140]. Notably, we did not observe tubulation of either marker in *piki-1(Q1507Stop)* mutants (S9 Fig), indicating that the tubulations detected with the dual-specificity marker are unlikely to be lysosomal in origin. We note, however, that NUC-1::mCherry–positive vesicles were more abundant but slightly smaller in *piki-1(Q1507Stop)* mutants, suggesting that PIKI-1 may directly or indirectly affect lysosomal compartments (S9D–S9G Fig). Our findings suggest that the tubulations observed with the AKT::oxGFP marker, as well as with TGN-38::GFP, may be extensions of sorting or recycling endosomal compartments, although they were not marked by any of our tested proteins.

### *nekl* defects may be suppressed by inhibition of PI(3,4)P$_2$ binding proteins

Our findings suggest that a reduction of PI(3,4)P$_2$ is responsible for the genetic suppression of *nekl-2; nekl-3* mutants by *piki-1* mutants. Moreover, a reduction of PI(3,4)P$_2$ would be expected to affect the binding of specific endocytic regulators to compartments including the early endosome. We therefore hypothesized that inhibition of one or more PI(3,4)P$_2$-binding proteins may contribute to the suppression of *nekl-2; nekl-3* mutants. To test this, we looked for potential PI(3,4)P$_2$-binding proteins involved in endocytic trafficking based on gene ontology terms. From this search, we identified two potential PI(3,4)P$_2$ interactors: F55D12.2, an ortholog of human SESTD1 (SEC14 and spectrin domain containing 1),

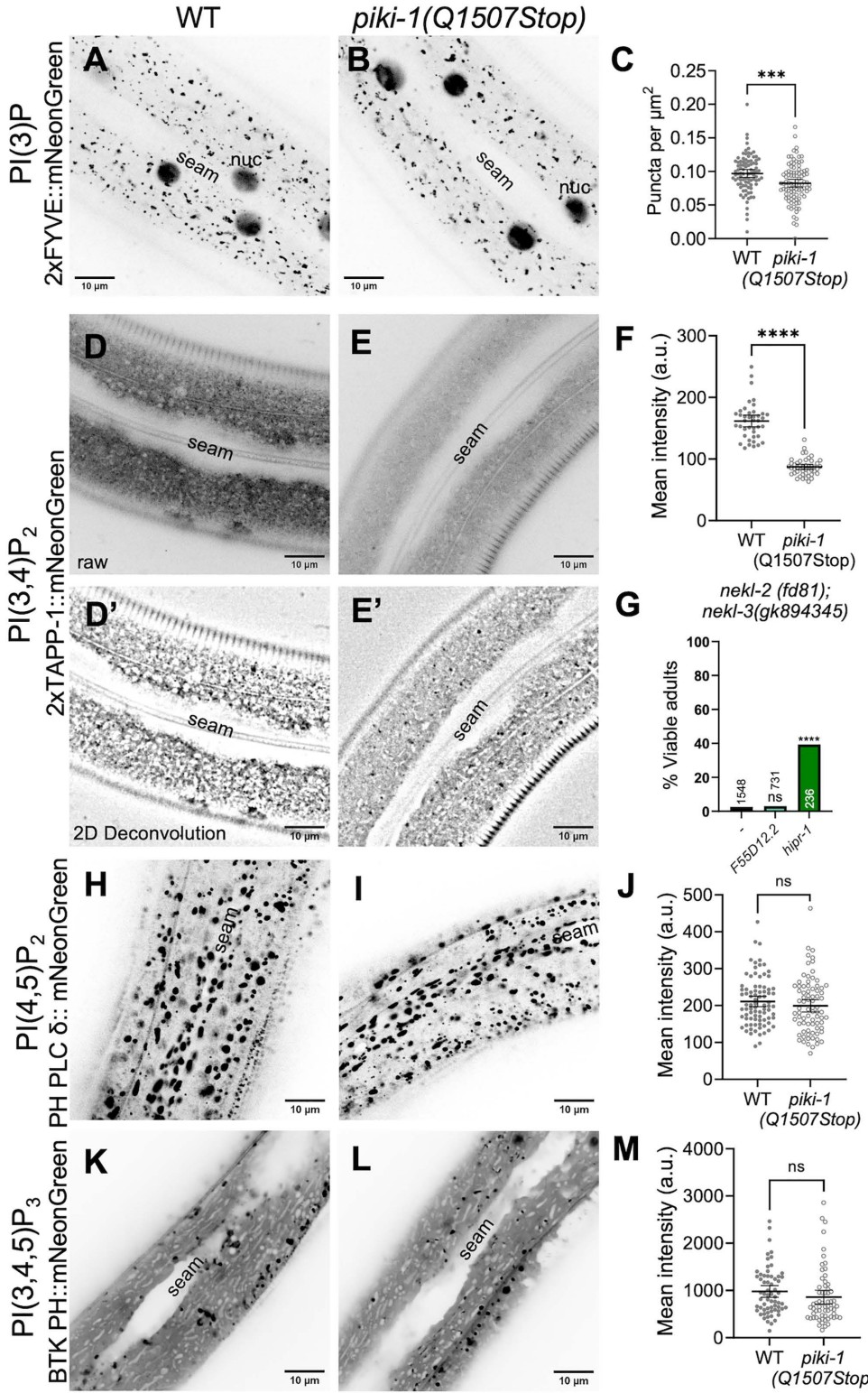

**Fig 6. Loss of PIKI-1 affects PI(3)P and PI(3,4)P₂ pools in the epidermis.** (A,B,D,E,H,I,K and **L**) Representative confocal images of day-1 adult worms expressing lipid sensors for **(A, B)** PI(3)P [P$_{hyp7}$::2xFYVE::mNeonGreen], **(D, E)** PI(3,4)P$_2$ [P$_{nekl-3}$::2xTAPP1::mNeonGreen], (H, **I)** PI(4,5) P$_2$ [P$_{hyp7}$::PH PLC δ::mNeonGreen], and **(K, L)** PI(3,4,5)P$_3$ [P$_{nekl-3}$::BTK PH::mNeonGreen] in (A, D, D' H and K) wild-type and (B,E, E', I and L)

*piki-1(Q1507Stop)* backgrounds. The seam cell is labeled in all images; example nuclei (nuc) are also indicated in panels A and **B.** (D', E') Representative confocal images of day-1 adult worms expressing P$_{nekl-3}$::2xTAPP1::mNeonGreen that have been processed using the 2D deconvolution algorithm in CellSens 4.2. **(C)** Puncta per unit area for worms expressing P$_{hyp7}$::2xFYVE::mNeonGreen. (F, J and **M)** The mean intensity for worms expressing **(F)** P$_{nekl-3}$::2xTAPP1::mNeonGreen, **(J)** P$_{hyp7}$::PH PLC δ::mNeonGreen, and **(M)** P$_{nekl-3}$::BTK PH::mNeonGreen. Dot plots show the mean and 95% CI. **(G)** The percentage of viable adults after injection of dsRNA for *F55D12.2* and *hipr-1* relative to control uninjected *nekl-2; nekl-3* worms. Statistical significance was determined using unpaired *t*-tests (C, F, J and M) or Fischer's exact test **(G)**. ****p ≤ 0.0001, ***p ≤ 0.001; ns, not significant. Raw data are available in S1 File.

which regulates lipid signaling [142], and HIPR-1, an ortholog of human HIP1 and HIPR1 (Huntingtin-Interacting Protein 1 [Related Protein]), proposed to regulate clathrin-mediated endocytosis and vesicular trafficking by linking endocytic proteins to the actin cytoskeleton [143]. Whereas inhibition of F55D12.2 by RNAi failed to suppress molting defects in *nekl-2; nekl-3* worms, possibly due to insufficient knockdown, *hipr-1(RNAi)* resulted in ~39% of *nekl-2; nekl-3* worms reaching adulthood (Fig 6G). Together, our findings are consistent with a model wherein reduced levels of PI(3,4)P$_2$, along with an accompanying reduction in the binding of PI(3,4)P$_2$-associated proteins such as HIPR-1, may contribute to the partial restoration of trafficking functions in *nekl* mutants and ultimately to the suppression of *nekl-2; nekl-3* molting defects.

## Discussion

Using a forward genetic approach to identify suppressors of *nekl*-associated molting defects, we isolated three independent alleles of the lipid modifier PIKI-1 and established a novel link between class II PI3K enzymes and NEKL protein kinases, which are essential for molting and epidermal membrane trafficking in *C. elegans*. We find that PIKI-1 regulates several endocytic processes in the epidermis, including a prominent role at early endosomes. PIKI-1 partially localizes to early endosomes, and its inhibition leads to a reduction in the size and number of membrane-bound structures marked with RAB-5 or EEA-1. Conversely, loss of PIKI-1 led to an expansion of the SNX-1–marked early endosome compartment, suggesting that PIKI-1 may be important for maintaining a balance between endosomal sub-compartments or microdomains. These phenotypes may be due to the combined effects of strongly reduced PI(3,4)P$_2$, modestly reduced PI(3)P, or increased PI(4)P, along with secondary effects on the lipid and protein composition of early endosomes in *piki-1* mutants. Notably, SNX1 and SNX-2 have a greater in vivo affinity for PI(3)P and PI(3,5)P$_2$ than for PI(3,4)P$_2$, potentially providing an explanation for the expansion of SNX-1 in *piki-1* mutants [106,107].

In addition to early endosomes, our expression studies indicated that PIKI-1 localizes to clathrin-coated pits, consistent with findings in mammalian cells [10,12,14,38,48,70]. Cell culture studies have shown that class II PI3Ks act at clathrin coated pits to produce PI(3,4)P$_2$, which then recruits proteins necessary for the completion of vesicle budding and the internalization of cargo [57,86]. However, PIKI-1 appeared to be largely dispensable for clathrin-mediated endocytosis in the *C. elegans* epidermis. This discrepancy may reflect differences in the mechanisms controlling endocytosis in polarized versus non-polarized cells or may be due to additional differences between cell types and species [144,145].

Using genetically encoded lipid-biosensors, we found that PIKI-1 is a major producer of PI(3,4)P$_2$ and a minor contributor to PI(3)P pools in hyp7. Both PI(3)P and PI(3,4)P$_2$ are found in association with early endosomes in the hyp7, consistent with a direct role for PIKI-1 in the production of these PPIns species. Other genetic data are consistent with PIKI-1 being the major producer of PI(3,4)P$_2$ in the epidermis. For example, we failed to observe genetic interactions when other pathway components that may contribute PI(3,4)P$_2$ synthesis were inhibited (*age-1* and *inpp-1*). However, these findings may reflect a failure to sufficiently reduce the expression of these genes, and we cannot rule out a role for alternative pathways in the production of PI(3,4)P$_2$ via the 5' dephosphorylation of PI(3,4,5)P$_3$ produced by Class I PI3Ks. In contrast, loss of PIKI-1 did not appear to affect PI(3,4,5)P$_3$ or PI(4,5)P$_2$ levels.

Reduction of PIKI-1 also resulted in hyper-tubulation defects based on a multi-specific lipid sensor (AKT-PH::GFP), as well as TGN-38, a trans-Golgi cargo that cycles through early endosomes, and LGG-1, a marker for phagosomes and

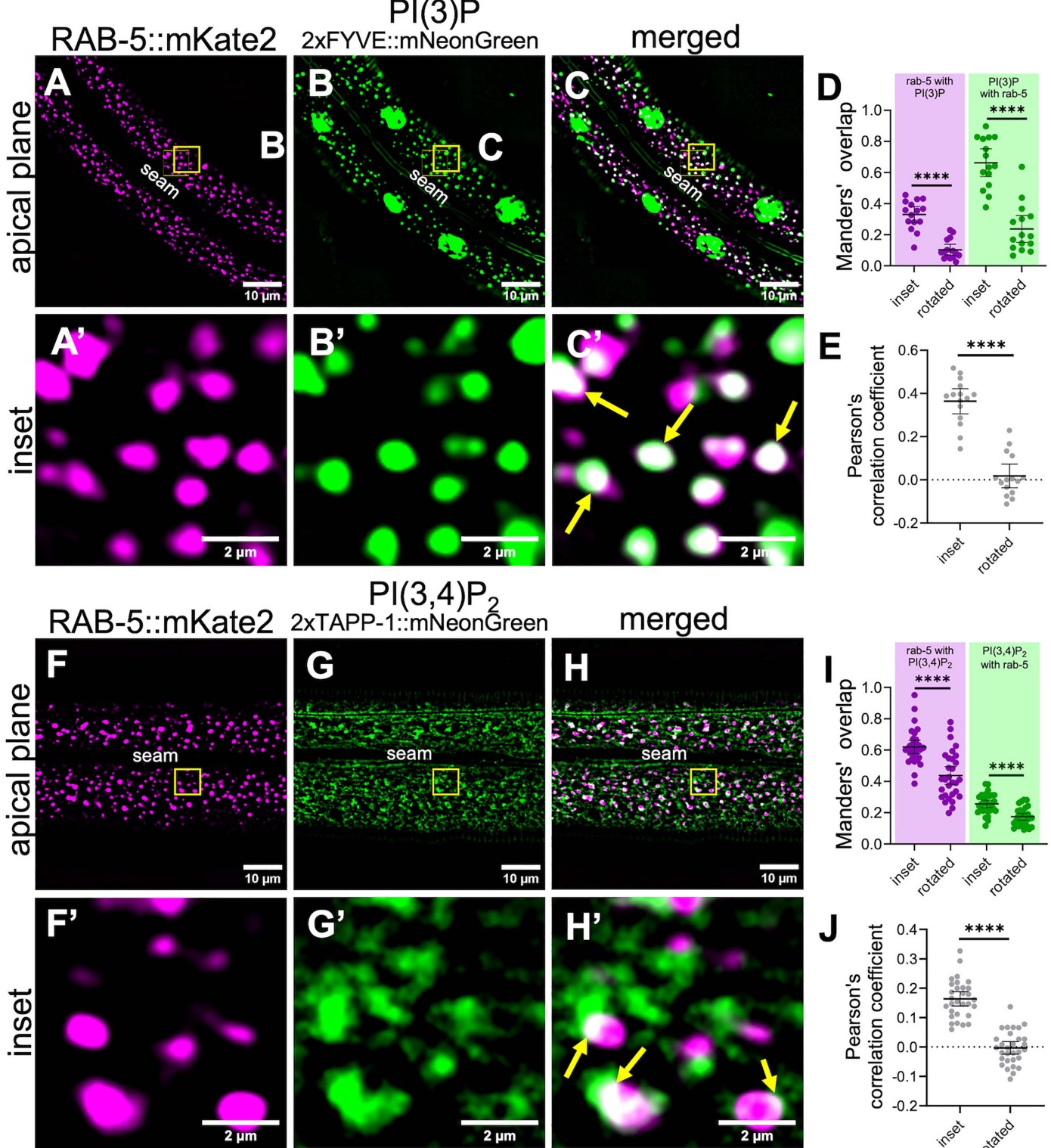

**Fig 7. PI(3)P and PI(3,4)P$_2$ localize at the early endosome.** (A–C' and F–H') Colocalization assays were carried out in worms expressing P$_{dpy-7}$:: mKate2::RAB-5 and (A–C') the PI(3)P lipid sensor [P$_{hyp7}$::2xFYVE::mNeonGreen] (n = 15) or (F–H') the PI(3,4)P$_2$ lipid sensor [P$_{nekl-3}$::2xTAPP1:: mNeonGreen] (n = 30). The seam cell is labeled **(A–H)**; yellow squares (A–C and F–H) indicate the enlarged insets (A'–C' and F'–H'). In merged insets

(C' and H'), yellow arrows indicate examples of colocalization. (D, E, I and **J**) Colocalization was quantified using Mander's overlap (D, I) and Pearson's correlation coefficient **(E and J)**. Dot plots show the mean and 95% CI. Statistical significance between rotated and inset values was determined using unpaired *t*-tests; ****p ≤ 0.0001. Raw data available in S1 File.

autophagosomes. Although lysosomes in *C. elegans* form elongated tubules during molting cycles [98], our studies did not support a major role for PIKI-1 in controlling the morphology of lysosomes. Moreover, lack of tubulations emanating from the Golgi (AMAN-2::GFP) in *piki-1* mutants suggests that the observed TGN-38 tubulations may arise from an early endosomal compartment. We speculate that loss of PIKI-1 may lead to defects in cargo sorting and/or endosomal tubule scission, leading to the observed tubules in *piki-1* mutants. Consistent with this idea, the $PI(3,4)P_2$ biosensor, 2xTAPP1, labeled shorter protrusions and tubule-like extensions associated with RAB-5 positive puncta.

Previous studies in *C. elegans* embryos and in the adult gonad indicate that PIKI-1 is required for the maturation and clearance of phagosomes following cell corpse engulfment [88–92]. Initially VPS-34 (a Class III PI3K), but not PIKI-1, was suggested to be the major producer of PI(3)P on embryonic phagosomes and autophagosomes [89]. However, other work indicated that PIKI-1, acts coordinately with VPS-34 and MTM-1 (a 3-phosphatase), to produce PI(3)P on embryonic phagosomes as well as in the germline (Fig 1B) [88,90,92]. Collectively these data suggest that the role of PIKI-1 in PI(3) production may be cell-type specific or confined to specific developmental time points.

In prior studies of the *nekls*, we identified several genetic suppressors with established roles in membrane trafficking including proteins controlling clathrin-mediated endocytosis, membrane lipid asymmetry, and endosomal-associated actin [74–76,78,79]. Other *nekl* suppressors appear to act independently of membrane trafficking per se and include regulators of cargo processing, cell signaling, and development [80,146–148]. Our current findings suggest that loss of PIKI-1 leads to the suppression of *nekl* molting defects through a mechanism that partially alleviates *nekl* trafficking defects at early endosomes. We propose that loss of PIKI-1 interferes with the functions of $PI(3,4)P_2$-binding proteins, such as HIPR-1, inhibition of which led to the suppression of *nekl-2; nekl-3* molting defects.

Broadly speaking, our results suggest that PIKI-1 and NEKL-2 may act in opposition given that loss of *piki-1* and depletion of NEKL-2::AID lead to apparent opposite effects on early endosomes [74]. Although our studies do not reveal a precise molecular connection, it is possible that PIKI-1 is a direct or indirect target for negative regulation by NEKL-2. Consistent with this, we recently identified PIKI-1 as a proximal interactor of NEKL-2 in the epidermis, raising the possibility that NEKL-2 may associate with and directly influence PIKI-1 activity [149]. More generally, our current and prior studies suggest that inhibition of the *nekls* leads to defective trafficking at multiple points within the endomembrane system, which lead to molting defects in part through the reduced uptake of sterol precursors required for molting gene transcription [73–75,78,79,147]. We further speculate that molting defects in *nekl* mutants can be partially alleviated by correcting for a subset of *nekl*-associated trafficking defects, consistent with our previous studies [74,75,78,79]. Future studies combining genetic, proteomic, and biochemical assays are expected to provide deeper insights into how conserved NIMA-related kinases regulate endocytic trafficking in conjunction with conserved trafficking components.

## Materials and methods

### Strains and propagation

All *C. elegans* strains were maintained per standard protocols and propagated at 22°C unless stated otherwise [150]. Strains in this study are listed in Table A in S2 File.

### RNAi

Standard dsRNA injection methods were used to conduct RNAi experiments [151]. Primers containing the T7 RNA polymerase–binding motif and corresponding to *piki-1, age-1, inpp-1, inpp-4b,* F55D12.2*,* and *hipr-1* were used to synthesize

dsRNA using the MEGAscript RNAi Kit (Invitrogen). dsRNA was injected at concentrations of 500–1000 ng/µL. Primer information is in Table B in S2 File.

## CRISPR mutant alleles

Alleles of *piki-1* (Q1507Stop) were created using established CRISPR-Cas9 protocols [152–154]. sgRNA and repair templates were synthesized by Integrated DNA technologies and Dharmacon-Horizon Discovery; ApE and CRISPRcruncher were used in the design of the guideRNA and repair templates [154,155]. Primer, sgRNA, and repair template sequences are provided in Table C in S2 File.

## Reporter strain construction

Plasmids for *C. elegans* epidermal-specific expression in the hyp7 syncytium used promoters for *semo-1*/Y37A1B.5 ($P_{hyp7}$) and *nekl-3* ($P_{nekl-3}$) as previously described [74,75,79]. We generated pDONR221 entry vectors containing coding regions for *C. elegans piki-1* (gift from Zheng Zhou, Baylor College of Medicine) [92], human *2x-TAPP1,* wherein the second repeat was codon optimized for *C. elegans* [156] and contained synthetic introns to promote expression (Integrated DNA Technologies); human *AKT* [139] and human *BTK* (Addgene Plasmid #51463) [157]. Cloning of the PH-domain from human AKT (pDONR221 AKT) into destination vector pCFJ1662 $P_{semo-1}$::GTWY::oxGFP::*let-858* (35G7) [74,125,126] was performed using the Gateway LR clonase II reaction (Invitrogen). *piki-1* and *BTK* pDONR221 clones were transferred into destination vector pCFJ1662 $P_{hyp7}$::GTWY::mNeonGreen::*let-858* (34H4) via the Gateway LR clonase II reaction. The *2X-TAPP1* pDONR221 clone was transferred into destination vector $P_{nekl-3}$::GTWY::mNeonGreen::*let-858* (pDF477, derived from pCFJ1662 (34H4)) [79] using the Gateway LR clonase II reaction. To generate single-copy integrations, standard miniMos procedures were followed [94]. In cases where we were unable to obtain integrations, we co-injected the expression clone and a plasmid containing *unc-119(+)* into a background of *unc-119(ed3)* and used the resulting stable extrachromosomal array in our analyses (*fdEx405, fdEx406,* S2 File).

## Auxin treatment

Indole-3-acetic acid (auxin) from Alfa Aesear was used to make a 100× stock auxin solution (0.4 M) by dissolving 0.7 g of auxin in 10 mL of 100% ethanol. For experiments, a mixture of 25 µL of stock auxin solution and 225 µL of autoclaved deionized water was added to NGM plates spotted with OP50 with day-1 adult worms present, at least 18 hours before imaging [75,94,158].

## Image acquisition

Fluorescence images were acquired using an Olympus IX83 P2ZF inverted microscope with a Yokogawa spinning-disc confocal head (CSU-W1). z-Stack images were acquired using a 100× 1.35 N.A. silicone oil objective. cellSens 4.2 software (Olympus Corporation) was used for image acquisition with a Hamamatsu-ORCA-Fusion camera. For each worm, z-stack slices were acquired every 0.2 µm for ~20 slices to encompass the epidermis of the worm.

## Image analysis to determine fluorescence, size, puncta per unit area, and percent positive pixels

All image analysis and quantification, was done using Fiji [159]. To quantify the mean intensity (measured in arbitrary units, a.u.), the intensity of the background of the image was first measured using the rectangle selection tool in an area of the image where there was no visible fluorescence. The resulting value was subtracted from the mean intensity value obtained from the epidermis (hyp7) of each animal in each picture, by using the polygon selection tool to select the appropriate region of interest (ROI) [74,79].

To quantify the average area of vesicles, number of puncta, or the percent positive pixels above threshold for a z-plane of interest which corresponds to a single slice at an appropriate plane from the obtained z-stacks, images underwent

processing to remove background. This was done through application of the rolling ball background subtraction method (in which background intensity values below the average within a 50-pixel radius surrounding a positive pixel are subtracted (Figs 3F, 3G, 3J and 3K 3N, 3O, 3R, 3S, and S9). For images for which rolling ball background subtraction did not work well, we used the minimum filter method (in which the central pixel is compared to other pixels within a 10-pixel radius and the minimum value in the window is replaced with the central pixel value to reduce noise (Figs 3B, 3C, S5A, S5B, S5D, S5E, S5G, S5H, S6A, S6B, S6E, S6F, S6I, S6J, S6M, and S6N). The filtered image was subtracted from the raw image using the image calculator function. After processing, all images were thresholded using the algorithm that worked best (or a manual threshold set using representative images; see S1 File for details and raw data). The "Despeckle" function was subsequently applied to all images to remove signal noise of ≤1pixel in size. Then the "Analyze Particles" function was applied to the processed images to determine the average area of vesicles. Within each experiment, the same background subtraction and threshold algorithms were used for all images. To account for variation in the size of the region of interest (ROI; drawn using the polygon tool to select hyp7 and exclude the seam cell) measured among worms, the number of puncta was divided by the area of the ROI (output in units of puncta per square micron).

### Image analysis to determine colocalization

All image analysis for colocalization was carried out on day-1 adults. For quantifying colocalization, the raw z-stack images were deconvoluted using the 2D deconvolution algorithm available in cellSens (ver. 4.2). The appropriate z-plane was then extracted from both the raw and deconvoluted images for each channel. To obtain a binary image to be used as a mask, deconvoluted images were thresholded. This binary mask was then combined with the raw image using the "AND" Boolean operation. The ROI was drawn around hyp7, excluding the seam cell, by using the polygon tool. To calculate the Pearson's correlation coefficient (R) and Mander's overlap (M) for these experiments, we used the BIOP JACoP plugin [160]. Merged images (containing both red and green channels) were used to determine the cases of significant overlap versus random co-occurrence. From these merged images, a small inset of 100 × 100 pixels (10,000 pixels$^2$) was sampled, and the R and M values were calculated using the BIOP JACoP plugin (shown as "inset" in the resulting graphs). To create a random distribution of green and red pixels of interest as a control against random coincidence, the red channel was rotated 90° in relation to the green channel using the transform function before the R and M values were calculated using the BIOP JACoP plugin (shown as "rotated") [79,161].

For convenience, some colocalization studies were carried out using strains that were trans-heterozygous for the tested markers (Figs 2H–L, 4, and S4). In the case of the PIKI-1::mNeonGreen and mKate2::RAB-5 markers, we noticed that homozygous worms exhibited ectopic expression of PIKI-1::mNeonGreen in the seam cell with strongly reduced expression in hyp7 (S10 Fig), suggesting a genetic interaction induced by the fluorescent tags or an overexpression artifact. Importantly, trans-heterozygous PIKI-1::mNeonGreen and mKate2::RAB-5 marked worms exhibited expression of PIKI-1::mNeonGreen specifically in hyp7, similar to the parental homozygous strains.

### Image analysis for phenotypes

Worms expressing P$_{hyp7}$::AKT::GFP were scored based on the appearance of the marker throughout the entire epidermis. Worms were considered to have diffuse expression if there was a uniform architecture of the GFP signal and the boundary between hyp7 and the seam cell was clear. Worms were considered to have aggregates if there were at least three large aggregations present within the z-stack that were not uniform in size or morphology and were distinct from the background and nuclear expression. Worms with tubulations had elongated compartments originating from two or more sites in hyp7. All worms expressing AKT::GFP that exhibited tubulations also contained aggregations.

Worms expressing TGN-38::GFP or mNeonGreen::LGG-1 were scored based on the absence or presence of tubulations within the epidermis. Tubulations were defined as being longer, extended vesicles or longer, thin protrusions extending through the epidermis.

## Statistical analysis

GraphPad Prism software was used to perform statistical tests in accordance with standard methods [162].

## Supporting information

**S1 Fig. RNAi of additional PPI modifiers.** (A) The proportion of viable progeny produced by *nekl-2(fd81); nekl-3(gk894345)* worms after injection by the indicated dsRNAs. (B) The proportion of viable progeny produced by *nekl-2(fd81); nekl-3(gk894345)* worms for the indicated genotype. Statistical significance was determined using Fischer's exact test; ****$p \le 0.0001$, ***$p \le 0.001$. Error bars indicate and 95% CI. Raw data available in this File.
(TIFF)

**S2 Fig. Endogenous CRISRP-Cas9 PIKI-1 expression.** (A, B) Representative confocal images of day-1 adults expressing CRISPR-tagged (A) PIKI-1::GFP and (B) PIKI-1::mScarlet. Note that accumulations present in the PIKI-1::mScarlet strain are likely the result of cleavage of mScarlet and its retention in lysosomes.
(TIFF)

**S3 Fig. PIKI-1 does not colocalize with late endosomes or lysosomes.** (A–C' and F–H') Colocalization of young adult worms expressing $P_{hyp7}$::PIKI-1::mNeonGreen with (A–C') GFP::RAB-7 and (F–H') NUC-1::mCherry. Yellow squares correspond to enlarged insets; yellow arrows indicate examples of colocalization (white). The seam cell is labeled in lower-magnification images. Colocalization was quantified using Mander's overlap (D, I) and Pearson's correlation coefficients (E, J). Dot plots show the mean and 95% CI. Statistical significance between rotated and inset values was determined using unpaired *t*-tests; ns, not significant. Raw data are available in S1 File.
(TIFF)

**S4 Fig. PIKI-1 does not colocalize with two recycling compartment markers.** (A–C' and F–H') Colocalization of young adult worms expressing heterozygous $P_{hyp7}$::PIKI-1::mNeonGreen with (A–C') heterozygous $P_{hyp7}$::mScarlet::RME-1 (n = 24) or (F–H') heterozygous CHAT-1::mKate (n = 20). Yellow squares correspond to the enlarged insets; yellow arrows indicate examples of colocalization (white). The seam cell is labeled in A–C and F–H. Colocalization was quantified using Mander's overlap (D, I) and Pearson's correlation coefficient (E, J). Dot plots show the mean and 95% CI. Statistical significance between rotated and inset values were determined using unpaired *t*-tests; ns, not significant. Raw data are available in S1 File.
(TIFF)

**S5 Fig. Effects of reduction of PIKI-1 function on clathrin and cargos.** (A, B, D, E, G and H) Representative confocal images of day-1 adults in wild-type and *piki-1(Q1507Stop)* mutants expressing (A, B) GFP::CHC-1, (D, E) mScarlet::CLIC-1, and (G, H) $P_{hyp7}$::LRP-1::GFP. The seam cell is labeled in all images. (C, F, and I) Mean intensities corresponding to markers shown in the left-hand panels; (J) Mean intensity for TGN-38::GFP in wild-type and *piki-1(Q1507Stop)* mutants (also see Fig 3M and 3N). Dot plots show the mean and 95% CI. Statistical significance was determined using unpaired *t*-tests; *$p \le 0.05$; ns, not significant. Raw data are available in S1 File.
(TIFF)

**S6 Fig. Effects of reduction of PIKI-1 function on additional endocytic compartments.** (A, B, E, F, I, J, M and N) Representative confocal images of day-1 adults in wild-type and *piki-1(Q1507Stop)* mutants expressing (A, B) GFP::RAB-7, (E, F) $P_{hyp7}$::GFP::RAB-11, (I, J) $P_{hyp7}$::mNeonGreen::RME-1, (M, N) $P_{hyp7}$::mNeonGreen::LGG-1. The seam cell is labeled in all images. Red arrows (N) indicate instances of tubulation. (C–P) Indicated parameters corresponding to the left-hand panels are indicated. Dot plots show the mean and 95% CI. Statistical significance (C–O) was determined

by an unpaired *t*-test or (P) Fisher's exact test. ****$p \leq 0.0001$; **$p \leq 0.01$; ns, not significant. Raw data are available in S1 File.
(TIFF)

**S7 Fig. Effects of *piki-1(ok2346)* deletion allele on PI(3)P and PI(4,5)P$_2$.** (A) Enlarged image of RAB-5::mKate and 2xTAPP-1::mNeonGreen [PI(3,4)P2] markers showing protrusions and tubule-like structures (green) emanating from early endosomes (magenta). (B, C, E and F) Representative confocal images of wild-type and *piki-1(ok2346)* day-1 adults expressing (A, B) a PI(3)P lipid sensor [P$_{hyp7}$::2xFYVE::mNeonGreen] and (D, E) a PI(4,5)P$_2$ lipid sensor [P$_{hyp7}$::PH PLC δ::mNeonGreen]. (D) Puncta per unit area for worms expressing P$_{hyp7}$::2xFYVE::mNeonGreen were slightly decreased. (G) Mean intensity for worms expressing P$_{hyp7}$::PH PLC δ::mNeonGreen was unchanged. Dot plots show the mean and 95% CI. Statistical significance was determined using unpaired *t*-tests; ****$p \leq 0.0001$; ns, not significant. Raw data are available in S1 File.
(TIFF)

**S8 Fig. Effects of PIKI-1 reduction of function on the multi-specific lipid sensor AKT-PH.** (A–C, E and F) Representative confocal images of day-1 adults expressing the PI(3,4)P$_2$/PI(3,4,5)P$_3$ lipid sensor P$_{hyp7}$::AKT::oxGFP in (A) wild-type, (B and C) *piki-1(Q1507Stop)*, or (E and F) *piki-1(ok2346)* backgrounds. White arrows (A) indicate nuclei. Orange arrows (B, E) indicate aggregations. Red arrows (C, F) indicate tubulations. (D, G) Individual worms expressing P$_{hyp7}$::AKT-PH::oxGFP were scored for the presence of diffuse labeling, aggregations, or aggregations and tubulations within the epidermis. Statistical significance of the differences in phenotype distributions were determined using Chi-squared tests; ****$p < 0.0001$. Raw data are available in S1 File.
(TIFF)

**S9 Fig. Effects of reduction of PIKI-1 function on lysosomal markers.** (A, B, D and E) Representative confocal images of day-1 adults expressing (A, B) P$_{hyp7}$::LMP-1::mNeonGreen or (D, E) NUC-1::mCherry in (A, D) wild-type and (B, E) *piki-1(Q1507Stop)* backgrounds. (C) Mean intensity was plotted for P$_{hyp7}$::LMP-1::mNeonGreen. (F) The number of puncta per unit area and (G) size of puncta were plotted for NUC-1::mCherry. Dot plots show the mean and 95% CI. Statistical significance was determined using unpaired *t*-tests; ****$p \leq 0.0001$; ns, not significant. Raw data are available in S1 File.
(TIFF)

**S10 Fig. Marker expression in worms homozygous for P$_{hyp7}$::PIKI-1::mNeonGreen and P$_{dyp-7}$::mKate2::RAB-5.** (A–C) Representative raw images of a day-1 adult homozygous for both P$_{hyp7}$::PIKI-1::mNeonGreen and P$_{dpy-7}$::mKate2::RAB-5. Both (A,B) single-channel and (C) merged images are shown. Note abnormal expression of PIKI-1 in the seam cell.
(TIFF)

**S1 Movie. P$_{nekl-3}$::2xTAPP1::mNeonGreen in the epidermis of a wild-type worm.** z-Stack movie showing the expression of the PI(3,4)P$_2$ sensor P$_{nekl-3}$::2xTAPP1::mNeonGreen from the apical to the basal plane in the epidermis of a wild-type worm.
(AVI)

**S2 Movie. P$_{nekl-3}$::2xTAPP1::mNeonGreen in the epidermis of a *piki-1(Q1507Stop)* worm.** z-Stack movie showing the expression of the PI(3,4)P$_2$ sensor P$_{nekl-3}$::2xTAPP1::mNeonGreen from the apical to the basal plane in the epidermis of a *piki-1(Q1507Stop)* mutant.
(AVI)

**S1 File. Raw data for manuscripts figures.** Excel file with individual labeled tabs containing raw data for Figs 1–7, S1, and S3–S9. Data are labeled according to figure numbers and panels.
(XLSX)

**S2 File. Supplemental Tables A-C.** Excel file with individual labeled tabs for Table A, strain list; Table B, RNAi sequences/reagents; and Table C, CRISPR sequences/reagents.
(XLSX)

## Acknowledgments

The authors thank Amy Fluet for editing this manuscript and the three anonymous reviewers for their valuable input, which improved the manuscript. The authors also thank Zheng Zhou (Baylor College of Medicine) for the plasmid construct containing *piki-1* cDNA and Reto Gassmann (University of Porto) for sharing GCP1474 ($P_{dpy7}::mKate-2::rab-5$) with us. Some strains were provided by the CGC, which was funded by the NIH Office of Research Infrastructure Programs (P40 OD010440). The authors acknowledge the Center for Advanced Scientific Instrumentation (CASI) at the University of Wyoming for access to spINBRE TIRF. This research was supported by an Institutional Development Award (IDeA) from the National Institute of General Medical Sciences of the NIH under Grant P20GM103432 and by NIH R35 GM136236 to DSF.

## Author contributions

**Conceptualization:** Gabrielle R. Reimann, Barth D. Grant, David S. Fay.

**Data curation:** Gabrielle R. Reimann, Philip T. Edeen, Sylvia Conquest, David S. Fay.

**Formal analysis:** Gabrielle R. Reimann, Philip T. Edeen, Sylvia Conquest, David S. Fay.

**Funding acquisition:** David S. Fay.

**Investigation:** Gabrielle R. Reimann, Philip T. Edeen, Sylvia Conquest, Barth D. Grant, David S. Fay.

**Methodology:** Gabrielle R. Reimann, Philip T. Edeen, Sylvia Conquest, Barth D. Grant, David S. Fay.

**Project administration:** David S. Fay.

**Resources:** Barth D. Grant, David S. Fay.

**Supervision:** David S. Fay.

**Validation:** Gabrielle R. Reimann, Philip T. Edeen, David S. Fay.

**Visualization:** Gabrielle R. Reimann, Philip T. Edeen, David S. Fay.

**Writing – original draft:** Gabrielle R. Reimann, Barth D. Grant, David S. Fay.

**Writing – review & editing:** Gabrielle R. Reimann, David S. Fay.

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
