## [Decision Letter · Decision Letter 0]

11 Jul 2025

PGENETICS-D-25-00615

PIKI-1, a class II phosphatidylinositol 3-kinase, functions in endocytic trafficking

PLOS Genetics

Dear Dr. David S. Fay,

Thank you for submitting your manuscript to PLOS Genetics. We apologise for the delay in the revision process due to the late receipt of the latest review. After careful consideration, we feel that it has merit but does not fully meet PLOS Genetics's publication criteria as it currently stands. Therefore, we invite you to submit a revised version of the manuscript that addresses the points raised during the review process.

Please submit your revised manuscript within 60 days. If you will need more time than this to complete your revisions, please reply to this message or contact the journal office at plosgenetics@plos.org. Please include the following items when submitting your revised manuscript:

We look forward to receiving your revised manuscript.

Kind regards,

Nathalie Pujol

Academic Editor

PLOS Genetics

Nathalie Pujol

Academic Editor

PLOS Genetics

Aimée Dudley

Editor-in-Chief

PLOS Genetics

Anne Goriely

Editor-in-Chief

PLOS Genetics

**Additional Editor Comments:**

additional minor comment: Could the authors comment on the ‘ectopic’ signal appearing in the seam of fig. S7? Is it related in any way to the signal in the seam in fig. 5H?

**Journal Requirements:**

At this stage, the following Authors/Authors require contributions: Philip T. Edeen, Sylvia Conquest, Barth D. Grant, and David S. Fay. Please ensure that the full contributions of each author are acknowledged in the "Add/Edit/Remove Authors" section of our submission form.

The list of CRediT author contributions may be found here: https://journals.plos.org/plosgenetics/s/authorship#loc-author-contributions

4) We notice that your supplementary Figures are included in the manuscript file. Please remove them and upload them with the file type 'Supporting Information'. Please ensure that each Supporting Information file has a legend listed in the manuscript after the references list.

5) We note that your Data Availability Statement is currently as follows: "All data is available in the supplementary files or available upon request.". Please confirm at this time whether or not your submission contains all raw data required to replicate the results of your study. Authors must share the “minimal data set” for their submission. PLOS defines the minimal data set to consist of the data required to replicate all study findings reported in the article, as well as related metadata and methods (https://journals.plos.org/plosone/s/data-availability#loc-minimal-data-set-definition).

- The points extracted from images for analysis..

6) Please ensure that the funders and grant numbers match between the Financial Disclosure field and the Funding Information tab in your submission form. Note that the funders must be provided in the same order in both places as well. State what role the funders took in the study. If the funders had no role in your study, please state: "The funders had no role in study design, data collection and analysis, decision to publish, or preparation of the manuscript.".

**Reviewers' comments:**

Reviewer's Responses to Questions

**Comments to the Authors:**

Reviewer #1: Reimann et al. present a rigorous and quantitative study demonstrating important roles of the class II PI3Kinase in endosomal organelles and cuticular molting. Using the power of genetics, they show that loss-of-function mutations in piki-1 suppress the molting defects of mutants that enlarge early endosomes, likely by decreasing the number and size of early endosomes. They also demonstrate that PI3,4P2 levels are altered in piki-1 mutants, suggesting that PIKI-1 is the major kinase acting on PI4P to generate PI3,4P2. This suggests the model that PI3,4P2 lipids influence early endosome size and number, which is further corroborated by the identification of a PI3,4P2-binding protein with similar genetic interactions to PIKI-1. These findings are a significant starting point for identifying how PI3,4P2 influences endosomal trafficking with players conserved across animals.

The authors started to explore the trafficking mechanism further and find that piki-1 mutations increase the number of tubules containing Golgi-PM-cycling TGN-38 and Atg8-like LGG-1. However, these markers are found on a diverse range of organelles, which makes it challenging to determine the step at which PIKI-1 or PI3,4P2 are likely to act. Their data suggests that the tubules are also likely covered in PIP3 and the authors propose that these are early endosomal tubules, but the nature and dynamics of the tubules are unclear due to the absence of two-color or time-lapse imaging. The tubulation findings identify a number of informative avenues to pursue in later research and expose gaping holes in our current knowledge of membrane trafficking, but are difficult to synthesize into a mechanistic model at this point.

Major comments

Line 194 - Do the authors think the premature stop codons in piki-1 are likely to cause nonsense-mediated decay or truncate the C2B domain? NMD could be tested experimentally by checking piki-1 mRNA levels, performing a PIKI-1 western (assuming antibodies exist), or suppressing NMD in their triple mutant strain. Also, it would be helpful to briefly comment on what the C2B domain is thought to do. That could help illuminate which subset of PIKI-1 functions are likely to contribute to nekl rescue.

Line 235 - Given the challenges in making PIKI-1 FP knock-ins with consistent localization patterns, do the authors know if any of their tagged PIKI-1s are functional? Does the mScarlet knock-in suppress nekl-2(-); nekl-3(-) defects like the other loss-of-function alleles? Does PIKI-1::mNG expression in hyp7 reinstate the growth phenotype in nekl-2(-); nekl-3(-); piki-1(-)? Also, could the authors comment on whether the hyp7 promoter is likely to cause PIKI-1 overexpression by comparing published levels of endogenous semo-1 and piki-1 mRNA in the hypodermis?

Line 366 - To demonstrate a direct role for PIKI-1 in early steps of autophagy, it would be necessary to show significant colocalization of PIKI-1 and LGG-1. This can be rephrased or addressed experimentally. Also, could the LGG-1+ tubules be a sign of CASM rather than autophagosomes? Testing that would require a fair amount of EM tomography to observe whether any tubular structures have single or double membranes. For this study, it may be better to give more open-ended interpretations.

Line 506 - Given that PIKI-1 is predicted to phosphorylate PI4P, piki-1 mutants may be predicted to have higher levels of PI4P (especially given the observed absence of an increase in PI4,5P2). Have the authors created any PI4P reporters, such as using the P4C domain of SidC? The authors should discuss the possibility that phenotypes could arise from altering PI4P turnover in addition to PI3,4P2 levels. Overexpressing a PI3,4P2 phosphatase or the PI5Kinase required to make PIP3 could also help distinguish between the roles of the different lipids.

A major challenge in studying trafficking is its cyclical nature, where it is difficult to tell if an increased number of tubules in a static image is due to increased formation of tubules or decreased vesicle release from tubules (longer-lived tubules). Can the author’s design any experiments to distinguish between increased tubulation and stalled tubules, i.e. time-lapse imaging or disrupting tubulation factors? Furthermore, comparing tubulation markers would be helpful to distinguish whether the tubulation of multiple organelles is altered or whether everything is likely to be happening on a Rab5+ early endosome.

Minor comments

Line 65 - PIP2 should be bisphosphate, not bi

Line 86 - “via PI(4,5)P2” may be getting into too much detail when introducing the formation of PI3,4P2 by class II PI3Ks. If the authors want to introduce PI4,5P2 conversion to PI4P, that would need a separate sentence, but it could also be omitted here for clarity on PI3Ks.

Fig. 1B - This diagram is very helpful, but would it be possible to move the PI & PI4P entries up to make it clear that Class II PI3Ks are not phosphorylating PI to PI4P? Especially after B&W printing, the green color code will be lost to the reader. With both class II and class III PI3Ks on the same arrow, it gets crowded if the vertical spacing is left symmetric as shown.

Line 165 - Missing a period before “The”

Line 196 - Can the authors speculate on why the S211N mutation would disrupt PIKI-1 function? Is it conserved among this class of PI3Kinases? Otherwise, it would be helpful to highlight in the text that it is in an uncharacterized domain if there is no published supporting data.

Line 277 - Localization is not equivalent to function, so the conclusion about localization data should be toned down to reflect similar localization between worms and mammals. These data raise the possibility that the function is also conserved, but “suggests the role is conserved” sounds too strong. Also, it would be helpful to add a sentence emphasizing that your data suggests PIKI-1 also localizes to unidentified discrete structures that are CLIC-1-negative and RAB-5-negative. These could be the LGG-1+ structures.

Line 290 - “only slightly altered” is vague. Could the authors describe more specifically what they observed?

Fig. 3A - Autophagosomes have double membranes, but the diagram shows a single membrane associated with LGG-1. Atg8 family proteins can also be conjugated to single membranes, aka CASM, LANDO, or LAP. Which do the authors think they are observing?

Fig. 3 Legend - lines 299 & 304 - RAB-7 is missing the FP used, presumably GFP::RAB-7.

Line 349 - It would be helpful to better distinguish between the Golgi markers when introducing AMAN-2, for example referencing what is known about the sub-Golgi localization of alpha-mannosidase II in mammalian cells and emphasizing that AMAN-2 is a Golgi resident protein. It would also be helpful to comment on the lack of AMAN-2+ tubules in piki-1 mutants to better contrast with the TGN-38 findings.

Fig. 4 - Do the authors think that piki-1 is specifically suppressing nekl-2 defects and not nekl-3 defects? The authors interpret these results in terms of organelle specificity, but is it possible to test suppression of individual Nekl phenotypes beyond organelle size?

Line 469 - missing 2 on PI3,4P2.

Line 481 - which movie?

Fig. 5 - Are PI4P or PI3,4P2 levels altered in nekl-2 or nekl-3 mutants?

Line 536-537 - provide full names for SESTD1 & HIP1 for better context

Line 538 - It’s important to repeat the caveat on RNAi when interpreting the negative result with F55D12.2 and contrasting it with hipr-1 RNAi. Otherwise, it gives the impression that F55D12.2 has been ruled out.

Line 565 - Given the AKT-PH tubules in piki-1 mutants, isn’t it more likely that either/both PI3,4P2 and PIP3 are enriched on tubules?

Reviewer #2: General Assessment

This study uses a forward genetic screen in C. elegans to reveal that loss-of-function mutations in piki-1, which encodes the class II PI3K PIKI-1, suppress molting defects associated with NEKL kinases. The authors identify PIKI-1 as a primary regulator of PI(3,4)P₂ in the epidermis and demonstrate that it controls endocytic trafficking by influencing early endosome morphology and protein recruitment. The work is founded on a strong experimental design, and the conclusions are generally well-supported by the data.

However, the manuscript's impact could be significantly enhanced with a more rigorous investigation of the underlying mechanisms, the inclusion of more quantitative analytical methods, and greater clarity in the presentation. The following revisions are recommended to strengthen the manuscript's claims.

Major Comments

1. The manuscript suggests that Class I PI3K is not involved in generating the relevant PI(3,4)P₂ pool. While the data are consistent with PIKI-1 being the primary source, the evidence presented may not be sufficient to definitively exclude any role for Class I PI3K. I recommend moderating this conclusion

2. Defining the Subcellular Localization of PIKI-1: The study localizes PIKI-1 to RAB-5/CLIC-1-positive early endosomes. However, relying on morphology and colocalization with only early endosome markers is insufficient to rule out its presence in other compartments, especially given the resolution of confocal microscopy. To more precisely define PIKI-1's site of action, I recommend expanding the colocalization analysis to markers for other key organelles in the endo-lysosomal pathway, such as: Late endosomes, Recycling endosomes, Lysosomes. This will provide a more comprehensive map of PIKI-1's localization and strengthen the claims about its specific role in early endocytic events.

3. A central conclusion is that piki-1 mutations impair the enzyme's phosphotransferase activity, leading to reduced PI(3,4)P₂. The current evidence is based on genetic suppression, which is indirect. The observed phenotypes in the truncated and missense mutants could result from protein instability rather than a specific defect in catalysis.

4. The study establishes a clear genetic relationship between PIKI-1 and NEKL, but the molecular nature of this interaction remains unclear. Is there any physical interaction between PIKI-1 and NEKL? Assess whether loss of piki-1 affects NEKL kinase activity or localization, which would help position the two proteins relative to one another in the signaling pathway.

5. The hypothesis that PI(3,4)P₂ reduction suppresses nekl defects via the effector protein HIPR-1 is compelling but requires more direct evidence. The logical chain connecting PIKI-1, PI(3,4)P₂, and HIPR-1 is currently incomplete.

Minors

1. Logical Flow & Focus: The transition to Class II PI3Ks (line 83) feels abrupt. Please add a sentence to frame why this class is understudied. I also suggest condensing the descriptions of Class I/III PI3Ks to expand on the known roles and open questions regarding Class II PI3Ks, better highlighting the novelty of your work.

Rationale: Please explicitly state the rationale for focusing on PIKI-1.

2, Citations issues

(Lines 82-87): For the role of PI(3,4)P₂ in endocytic recycling, please cite more direct evidence, such as Roman-Fernandez et al. (ref 51).

(Lines 67-71): If available, add key foundational references regarding PI(4,5)P₂'s role in clathrin-mediated endocytosis.

3. (Line 326): The data show changes in endosome morphology, not function. Rephrasing "role in the function of early endosomes" to "role in the biogenesis or homeostasis of early endosomes" would be more accurate.

(Line 335): The term "only a minor or indirect role" is too definitive. Removing "only" creates a more appropriately cautious statement.

(Line 291): Remove "largely" for a more direct statement.

(Line 363-364): The phrase "although the effect on LGG-1 was less robust" is an unsupported comparison and should be removed or statistically substantiated.

4. Fig S7: The nNeonGreen signal driven by the Phyp7 promoter appears in seam cells. Please clarify if this is expected and whether it might affect colocalization analysis.

5. Please standardize the font sizes of labels within figures for consistency and a more professional appearance (e.g., Fig 3 H, I, L, M; Fig S3 A-F).

6. Please add the full strain names, in addition to allele names to ensure reproducibility.

Reviewer #3: Summary:

Two conserved NIMA127 related Ser/Thr protein kinases, NEKL-2 (NEK8/9 in mammals) and NEKL-3 (NEK6/7), and a number of their binding partners, are essential for proper molting via the regulation of endocytic trafficking. In nekl-2; nekl-3 double mutants, molting is arrested. These mutants suffer 98% larval lethality. Through a well-designed forward genetic screen, the authors isolated a number of extragenic suppressor mutants of the synthetic lethal phenotype. Among the suppressor mutants there are three loss-of-function alleles of piki-1, which encodes the only Class II PI-3 kinase in C. elegans. This manuscript reports the function of PIKI-1 in endocytic recycling in hypodermal cells. The authors found that the primary activity of PIKI-1 is to produce PI(3,4)P2 on the surfaces of early endosomes. Consistent with this activity, PIKI-1 is localized to clathrin-associated compartments and to early endosomes. In piki-1(lof) mutants, among a number of endocytic-related vesicles, early endosomes are specifically abnormal. Particularly, there are fewer early endosomes, and these endosomes are smaller. There are also other phenotypes related to endosomes, such as the tubulation phenotype. Regarding the functional relationship between PIKI-1 and NEKL-2/-3, the suppression of the lethal phenotype displayed by the nekl-2; nekl-3 double mutants by the lof mutations of piki-1 indicates that PIKI-1 acts downstream of NEKL-2 and NEKL-3 and antagonizes their functions. In support of this conclusion, piki-1 mutations appear to suppress the early endosome number and morphological defects displayed by nekl-2 mutants (yet this reviewer has some questions regarding the impact of the opposite early endosome defects displayed by piki-1 and nekl-2 single mutants (see below – Major Concerns)). Furthermore, inactivation of HIPR-1, a potential PI(3,4)P2-binding protein, partially suppresses the larval lethality of the nekl-2; nekl-3 double mutants. This result support the model that “reduced levels of PI(3,4)P2 on early endosomes, along with an accompanying reduction of PI(3,4)P2-binding proteins such as HIPR-1, lead to the suppression of nekl-2; nekl-3 molting defects and to the partial restoration of trafficking functions”.

Evaluation:

This manuscript reveals, for the first time, the potential function of PI(3,4)P2 in an essetial developmental process. It also reveals that the primary function of PIKI-1 is to produce PI(3,4)P2 on early endosomes. Unlike Classes I and III PI-3 kinases, Class II PI-3 kinases are not as well characterized. Class II PI-3 kinases are implicated in vesicle trafficking, cell signaling, and cell migration events. C. elegans PIKI-1 is known to facilitate the degradation of engulfed apoptotic cells confined inside phagosomes by producing PI(3)P in addition to Class III PI-3 kinase VPS-34. The report that PIKI-1’s function in early endosomes by the authors further discovers additional functions of PIKI-1 in cellular functions. The suppression of nekl-2; -3 double mutants further discloses the function of PIKI-1 in development. These novel discoveries should be shared with researchers studying phospholipid, intracellular trafficking, cell signaling, and animal development. This manuscript will be interested by a broad audience. That said, there are a few weak or unclear aspects that needs clarification and/or improvement. These aspects are described below.

Major concerns:

1. The molecular mechanisms behind the functional relationship between the NIMA127 related Ser/Thr protein kinases NEKL-2 and -3 and PI-3 kinase PIKI-1:

The suppression effects of PIKI-1 strongly suggest that NEKL-2/-3 act to inhibit the activity of PIKI-1. what is the molecular mechanism behind this inhibitory effect? This is a challenging question and there are many aspects to explore. For one of those aspects, which is to determine whether this inhibitory functions of the NEKL proteins targets PI(3,4)P2 on early endosomes, the authors should test:

(1) in nekl-2/-3 mutants, whether the level of PI(3,4)P2 increases on early endosomes.

If NEKL-2/-3 act as inhibitors of the PI-3 kinase activity of PIKI-1, one would expect that inactivating these upstream inhibitors will result in more PIKI-1 products (PI(3,4)P2) on early endosomes.

(2) whether HIPR-1 binds PI(3,4)P2.

HIPR-1 is a proposed PI(3,4)P2-binding protein. However, whether this is the case has not been tested. The HIPR-1/PI(3,4)P2 binding ability needs to be tested in vitro and/or in vivo, in order to determine whether the proposed model is correct.

2. How are the defects in the morphology and amount of early endosomes related to the developmental defects.

-- In piki-1(lof) mutants, the authors report that early endosomes are smaller and fewer in number than wild-type. In nekl-2(-) mutants, the authors have observed that early endosomes are bigger and more in number than wild-type, and the early endosomal report displays a brighter intensity. Some clarification is needed as to which (or both) of the opposite phenotypes displayed by the piki-1(-) and nekl-2(-) single mutants will impair the efficiency of endosomal trafficking. Does the presence of larger and more early endosomes indicate an arrest of the proper processing of early endosomes? Does the reduction of early endosome size and number indicate a reduction of early endosome biogenesis? The impacts of these anormalies have not been discussed. Also, it seems that, although piki-1(-) suppresses both the endosomal morphology and amount defects and the larval lethality, this report still has not firmly established that early endosomal defects is the cause for the molting defect of the nekl-2; -3 mutants. The authors need to discuss these issues and other possible models.

3. The molting defect occurs during larval development, yet the characterization of PIKI-1 subcellular localization and the effect of the piki-1 mutations in the signal intensity of a number of fluorescent reporters were performed in one-day old adults. What are the results of the cell biological assays in larval stages? Are they the same as what have been observed in adults?

Specific concern:

Discussion, lines 590-594:

“Previous studies in C. elegans embryos and in the adult gonad indicate that PIKI-1 has a primary role in autophagy in these contexts [72,74–76]. For example, PI(3)P production, which controls autophagy in the C. elegans embryo, is dependent on the coordination among PIKI-1, VPS-34, and MTM-1 (a 3-phosphatase). In addition, loss of PIKI-1 in the embryo and in the gonad strongly impairs autophagosome clearance [76]”.

--None of references 72, 74, 75, 76 indicates that PIKI-1 plays a primary role in autophagy. Specifically, reference 76 reports that loss of PIKI-1 function strongly impairs the clearance of phagosomes that contain apoptotic cells, not autophagosomes. The authors seem to mix up autophagosomes, which are ER-originated intracellular organelles, with phagosomes, which are a vacuole produced by engulfment.

**Have all data underlying the figures and results presented in the manuscript been provided?**

Reviewer #1: Yes

Reviewer #2: Yes

Reviewer #3: Yes

PLOS authors have the option to publish the peer review history of their article (what does this mean? ). If published, this will include your full peer review and any attached files.

**Do you want your identity to be public for this peer review?** For information about this choice, including consent withdrawal, please see our Privacy Policy .

Reviewer #1: No

Reviewer #2: No

Reviewer #3: No

**Figure resubmission:**
---

## [Decision Letter · Decision Letter 1]

6 Feb 2026

Dear Dr Fay,

We are pleased to inform you that your manuscript entitled "PIKI-1, a class II PI 3-kinase, functions in endocytic trafficking" has been editorially accepted for publication in PLOS Genetics. Congratulations!

Yours sincerely,

Nathalie Pujol

Academic Editor

PLOS Genetics

Nathalie Pujol

Academic Editor

PLOS Genetics

Aimée Dudley

Editor-in-Chief

PLOS Genetics

Anne Goriely

Editor-in-Chief

PLOS Genetics

BlueSky: @plos.bsky.social

Comments from the reviewers (if applicable):

The authors should carfully review Figure S9B&C

Reviewer's Responses to Questions

**Comments to the Authors:**

Reviewer #1: The authors have significantly improved the manuscript and include exciting new data that supports their hypotheses, such as a catalytically inactive piki-1 mutant.

The addition of SNX-1 localization is impactful, but underdiscussed. I think Fig. S7 should be added to a main figure. SNX-1 is associated with recycling tubules and retromer trafficking, but these are not discussed in the manuscript. The increase in SNX-1 labeling is consistent with the author’s observations of a shift in sorting endosome domain identity and increased tubules (although the dramatic TGN-38/LGG-1/AKT PH tubules remain mysterious).

The helpful diagram in Fig. 3A also needs SNX-1 added to a sorting endosome tubule.

Fig. S7A, Scarlet is misspelled

Reviewer #3: The authors addressed all of my concerns by explanation and discussion. In a particular case, although the authors did not perform experiments to examine whether the level of PI(3,4)P2 increases on early endosomes, which is expected if NEKL-2/-3 act as inhibitors of the PI-3 kinase activity of PIKI-1, the authors did explain the time constrain. They also provided new experimental evidence indicating that the NEKL-2 and PIKI-1 might directly or indirectly interact in a protein complex. I am thus satisfied by the authors revision.

**Have all data underlying the figures and results presented in the manuscript been provided?**

Reviewer #1: Yes

Reviewer #3: Yes

PLOS authors have the option to publish the peer review history of their article (what does this mean? ). If published, this will include your full peer review and any attached files.

**Do you want your identity to be public for this peer review?** For information about this choice, including consent withdrawal, please see our Privacy Policy .

Reviewer #1: No

Reviewer #3: No

**Data Deposition**

http://datadryad.org/submit?journalID=pgenetics&manu=PGENETICS-D-25-00615R1

**Press Queries**

---

## [Editor Report · Acceptance letter]

PGENETICS-D-25-00615R1

PIKI-1, a class II PI 3-kinase, functions in endocytic trafficking

Dear Dr Fay,

We are pleased to inform you that your manuscript entitled "PIKI-1, a class II PI 3-kinase, functions in endocytic trafficking" has been formally accepted for publication in PLOS Genetics! Your manuscript is now with our production department and you will be notified of the publication date in due course.

With kind regards,

Anita Estes

PLOS Genetics

On behalf of:
